# Multi-year Downscaling Application of Two-Way Coupled WRF3.4 and CMAQv5.0.2 over East Asia for Regional Climate and Air Quality Modeling: Model Evaluation and Aerosol Direct Effects

Chaopeng Hong[1,2,3], Qiang Zhang[1,4], Yang Zhang[2,4], Youhua Tang[5,6], Daniel Tong[5,6,7], and Kebin He[1,3,4]

[1]Ministry of Education Key Laboratory for Earth System Modeling, Department of Earth System Science, Tsinghua University, Beijing 100084, China
[2]Department of Marine, Earth, and Atmospheric Sciences, North Carolina State University, Raleigh, NC 27695, USA
[3]State Key Joint Laboratory of Environment Simulation and Pollution Control, School of Environment, Tsinghua University, Beijing 100084, China
[4]Collaborative Innovation Center for Regional Environmental Quality, Beijing 100084, China
[5]Cooperative Institute for Climate and Satellites, University of Maryland, College Park, Maryland, MD 20740, USA
[6]Center for Spatial Information Science and Systems, George Mason University, Fairfax, VA 22030, USA
[7]NOAA Air Resources Laboratory, 5830 University Research Court, College Park, Maryland, MD 20740, USA

*Correspondence to*: Qiang Zhang (qiangzhang@tsinghua.edu.cn) and Yang Zhang (yzhang9@ncsu.edu)

**Abstract.** In this study, a regional coupled climate-chemistry modeling system using the dynamical downscaling technique was established by linking the global Community Earth System Model (CESM) and the regional two-way coupled Weather Research and Forecasting - Community Multiscale Air Quality (WRF-CMAQ) model for the purpose of comprehensive assessments of regional climate change and air quality and their interactions within one modeling framework. The modeling system was applied over East Asia for a multiyear climatological application during 2006-2010 driven with CESM downscaling data under Representative Concentration Pathway 4.5 (RCP 4.5) as well as a short-term air quality application in representative months in 2013 driven with a reanalysis dataset. A comprehensive model evaluation was conducted against observations from surface networks and satellite observations to assess the model's performance. This study presents the first application and evaluation of the two-way coupled WRF-CMAQ model for climatological simulations using the dynamical downscaling technique. The model was able to satisfactorily predict major meteorological variables. The improved statistical performance for the 2-m temperature (T2) in this study (with a mean bias of -0.6 ℃) compared with the Coupled Model Inter-comparison Project Phase 5 (CMIP5) multi-models might be related to the use of the regional model WRF and the bias-correction technique applied for CESM downscaling. The model showed good ability to predict $PM_{2.5}$ in winter (with a normalized mean bias (NMB) of 6.4% in 2013) and $O_3$ in summer (with an NMB of 18.2% in 2013) in terms of statistical performance and spatial distributions. Compared with global models that tend to underpredict $PM_{2.5}$ concentrations in China, WRF-CMAQ was able to capture the high $PM_{2.5}$ concentrations in urban areas. In general, the two-way coupled WRF-CMAQ model performed well for both climatological and air quality applications. The coupled modeling system with direct aerosol feedbacks predicted aerosol optical depth relatively well and significantly reduced the overprediction in downward shortwave radiation at the surface (SWDOWN) over polluted regions in China. The

performance of cloud variables was not as good as other meteorological variables, and underpredictions of cloud fraction resulted in overpredictions of SWDOWN and underpredictions of shortwave and longwave cloud forcing. The importance of climate-chemistry interactions was demonstrated via the impacts of aerosol direct effects on climate and air quality. The aerosol effects on climate and air quality in East Asia (e.g., SWDOWN and T2 decreased by 21.8 W m$^{-2}$ and 0.45 ℃, respectively, and  most pollutant concentrations increased by 4.8%~9.5% in January over China's major cities) were more significant than in other regions because of higher aerosol loadings that resulted from severe regional pollution, which indicates the need for applying online-coupled models over East Asia for regional climate and air quality modeling and to study the important climate-chemistry interactions. This work established a baseline for WRF-CMAQ simulations for a future period under the RCP4.5 climate scenario, which will be presented in a future paper.

## 1 Introduction

Climate change and air pollution are two critical environmental issues that humanity must face. There are complex interactions between air pollution and climate change (Fiore et al., 2012; von Schneidemesser et al., 2015; Fuzzi et al., 2015). Air pollutants (e.g., aerosols) have direct effects on radiative forcing by scattering or absorbing incoming radiation and also indirect effects via their role in cloud formation; the effects in turn affect climate systems. Climate change can affect meteorological fields (e.g., temperature, humidity, precipitation, wind speed, cloud cover, and boundary layer mixing) as well as natural emissions (e.g., biogenic volatile organic compounds (BVOCs) emissions, soil and lightning nitrogen oxides (NO$_x$) emissions, and dust emissions) and thereby affect air quality. Global climate and chemistry modeling simulations (Fiore et al., 2012; Stevenson et al., 2013; Kim et al., 2015) have been conducted to investigate global climate change and air quality under the Special Report on Emissions Scenarios (SRES) and the Representative Concentration Pathways (RCPs) scenarios developed for the Intergovernmental Panel on Climate Change (IPCC) Fourth Assessment Report (AR4) (IPCC, 2007) and Fifth Assessment Report (AR5) (IPCC, 2013). Global climate models used in the AR5 include more detailed representations of aerosol and cloud processes and their interactions than those used in the AR4. There is high confidence in the radiative forcing mechanisms due to aerosol–radiation interactions, although low confidence in the forcing mechanisms of aerosol–cloud interactions in the models remains (IPCC, 2013).

However, global climate/chemistry models applied at a coarse spatial resolution may not well resolve mesoscale features over a regional domain of interest or well predict local air quality and thus are not suitable for high-resolution regional climate, air quality and health impact studies. As a result of these deficiencies, the dynamical downscaling technique has been widely used in regional climate studies (Oh et al., 2014; Wang et al., 2015; Xu et al., 2015). Dynamical downscaling uses initial conditions (ICs) and boundary conditions (BCs) from global models to drive regional models for high-resolution simulations. Several regional air quality studies using dynamical downscaling approaches have been conducted to predict future air quality under a changing climate at a regional scale (Gao et al., 2013; Penrod et al., 2014; Sun et al., 2015). However, these studies tended to use offline models—chemical transport models (CTMs) driven by future climate archived

from general circulation models (GCMs), lacking climate-chemistry interactions. Previous regional downscaling studies tended to focus on extreme climate events or the impacts of climate change on air quality rather than on the important chemistry and climate interactions.

Online-coupled regional climate-chemistry models have developed rapidly in recent years (Zhang et al., 2008; Baklanov et
al., 2014). Recently, the online-coupled Weather Research Forecast (WRF) model with Chemistry (WRF/Chem) was evaluated for decadal application over the continental U.S. under RCP 4.5 and RCP 8.5 (Yahya et al., 2016, 2017a) and applied to decadal projections of future climate and air quality under both scenarios (Yahya et al., 2017b). The Community Multi-scale Air Quality (CMAQ) model has historically been an offline model developed by the U.S. Environmental Protection Agency (EPA) and widely used for air quality simulations over numerous countries and regions (Wang et al.,
2009; 2012; Liu et al., 2010; Gao et al., 2013; Penrod et al., 2014; Sun et al., 2015; Zheng et al., 2015; Hu et al., 2016); it has recently been further developed to provide an online-coupled version with the Weather Research Forecast (WRF) model to simulate feedbacks between chemistry and meteorology (Wong et al., 2012). Several applications of the two-way coupled WRF-CMAQ model have been conducted to evaluate the performance of the coupled model system and to investigate aerosol direct effects (Wang et al., 2014; Gan et al., 2015; Hogrefe et al., 2015; Xing et al., 2016) and indirect effects (Yu et
al., 2014) on climate and air quality. However, there is a lack of comprehensive evaluation of the two-way coupled WRF-CMAQ model over East Asia, where aerosol loadings are extremely high and have been found to have great impacts on regional climate and air quality (Wang et al., 2014; Liu et al., 2016). Additionally, the two-way coupled WRF-CMAQ model has not been applied and evaluated for multi-year climatological modeling.

In this study, following the work of Yahya et al. (2016, 2017a and b), a regional coupled climate-chemistry modeling system
using the dynamical downscaling technique was established for the purpose of comprehensive assessments of regional climate change and air quality and their interactions within one modeling framework (see Figure 1). The two-way coupled WRF-CMAQ model, which takes into account the air quality and climate interactions, is driven by the Community Earth System Model (CESM) implemented with advanced chemistry and aerosol treatments by North Carolina State University (NCSU) (hereafter CESM-NCSU) (He and Zhang, 2014; He et al., 2015a, b; Gantt et al., 2014; Glotfelty et al., 2017a, b;
Glotfelty and Zhang, 2017) for high-resolution regional simulation under a changing climate. Both meteorological dynamical downscaling and chemical composition downscaling from the CESM-NCSU were applied following the work of Yahya et al. (2016, 2017a and b). The dynamical downscaling methods fully take advantage of global climate-chemistry models that can well predict large-scale global changes and regional models that can better represent regional phenomena. The modeling system was applied over East Asia for a climatological application driven with CESM downscaling data for 5 years from
2006 to 2010 under RCP 4.5 as well as an air quality application in 2013 driven by the National Centers for Environmental Prediction (NCEP) Final Reanalyses (NCEP-FNL) dataset. Comprehensive model evaluation for meteorological, chemical and aerosol-cloud-radiation variables was conducted against surface observations and satellite observations to assess and improve the model's performance for regional climatological and air quality applications. This study presents the first

application and evaluation of the two-way coupled WRF-CMAQ model for climatological-type simulations using the dynamical downscaling technique; it also demonstrates the importance of climate-chemistry interactions via the impacts of aerosol direct effects on climate and air quality. The main goals of this work are to evaluate the WRF-CMAQ's capability in reproducing the observations and to establish a baseline simulation during a current year period, which will be compared with a simulation during a future year period in order to assess the impacts of changes in climate and emissions on future air quality over East Asia in future work.

This paper is organized as follows. Section 2 describes the model configurations and simulation design, dynamical downscaling methods and evaluation protocols. Section 3 presents the results of comprehensive model evaluations for climatological and air quality applications, the improvements of model performance within the modeling system, and aerosol direct effects on regional climate and air quality. Section 4 summarizes the major conclusions and limitations of this study.

## 2 Model setup and evaluation protocol

### 2.1 Model configurations and simulation design

The two-way coupled WRF-CMAQ (WRF3.4 and CMAQv5.0.2) model is used for regional climate and air quality simulations. More details of the two-way coupled WRF-CMAQ are described by Wong et al. (2012). The current release of the WRF-CMAQ model supports the Rapid and accurate Radiative Transfer Model for General Circulation Models (RRTMG) radiation scheme for shortwave aerosol direct effects and uses a core-shell model to perform the aerosol optics calculation. It does not include aerosol indirect effects that result from interactions between aerosols and cloud microphysics. The detailed model configurations for the climatological application in this study are shown in Table 1. The WRF model configuration included the Morrison double-moment scheme (Morrison et al., 2009), version 2 of the Kain-Fritsch cumulus scheme (Kain, 2004), the Asymmetric Convective Model version 2 (ACM2) planetary boundary layer (PBL) scheme (Pleim, 2007), the Pleim–Xiu land surface model (Xiu and Pleim, 2001), and the RRTMG shortwave and longwave radiation scheme (Iacono et al., 2008). The CMAQ model was configured using the Carbon Bond 2005 (CB05) chemical mechanism (Yarwood et al., 2005; Whitten et al., 2010) and the sixth generation CMAQ aerosol module (AERO6) (Appel et al., 2013). The regional domain using a horizontal resolution of 36 km covered most of China and parts of East Asia. The two-way coupled WRF-CMAQ used the same vertical resolution for WRF and CMAQ, i.e., 23 sigma layers from the surface to 100 hPa.

Several modifications in model inputs and treatments were made in this study to improve the model performance. These included (1) correcting the surface roughness by increasing the surface drag (which is applied to the friction velocity) by 1.5 times when calculating wind speeds in the ACM2 PBL scheme to reduce the overpredictions of wind speeds, which are likely caused by low surface drag due to the inappropriate representation of surface roughness because the detailed surface structure cannot be reproduced at a coarse grid resolution of 36-km (Mass and Ovens, 2010; Zheng, et al., 2015; Zhang et al.,

2016b); (2) using the inline Biogenic Emissions Inventory System (BEIS3) model (Vukovich and Pierce, 2002; Schwede et al., 2005) over East Asia; (3) revising the default dust module developed by Tong et al. (2017) with updated friction velocity thresholds to generate more dust emissions following the work of Dong et al. (2015); (4) using bias-corrected chemical boundary conditions (BCs)/initial conditions (ICs) from CESM rather than using the fixed BCs/ICs provided by the operational CMAQ system.

Table 2 shows the four simulations conducted in this study. Climatological application (CESM_BASE) was driven by the climatological dataset (CESM-NCSU) over a 5-yr period (2006-2010) and aimed to assess the model performance on a climatological average timescale. Air quality application (NCEP_BASE) was driven by a reanalysis dataset (NCEP-FNL, NCEP Final Reanalysis) and aimed to assess the model performance for short-term air quality application. The air quality application was conducted for three representative months (January, April, and July) in 2013 because more surface air quality monitoring data were available for the evaluation of chemical predictions (refer to Section 2.4). NCEP_BASE_WoImp simulation without the improvements indicated above was designed for comparison to support the improvements made in NCEP_BASE. Sensitivity simulation without aerosol feedback (CESM_BASE_Sens) was designed to assess the aerosol direct effects on regional climate and air quality.

In order to simulate regional meteorology as accurately as possible and preserve the chemistry–meteorology feedbacks, re-initialization in WRF was used in the multi-year climatological application. The climatological simulations were reinitialized every 15 days in this work, which provides a compromise to allow the simulation of mesoscale features and aerosol feedbacks while periodically constraining the meteorological fields not significantly deviated from the GCM. Qian et al. (2003) found that frequent re-initialization with frequencies of 10 days to 1 month improved the accuracy in regional climate downscaling.

## 2.2 Dynamical downscaling from CESM-NCSU

The CESM-NCSU model with advanced chemistry and aerosol treatments has been applied for decadal global climate and air quality predictions to simulate the "current" climate scenario (2001–2010) and the "future" climate scenario (2046-2055) driven with the RCPs emissions (projected from base year 2000) (Glotfelty et al., 2017a and Glotfelty and Zhang, 2017). The CESM simulation for 2001–2010 is performed with fully coupled CESM1.2.2 with the B_2000_STRATMAM7_CN configuration, which includes prognostic simulation of the atmosphere, ocean, land, and sea ice from the various component models. The initial conditions for CAM5.1 are derived from a 10 year (1990–2000) CAM5.1 standalone simulation with the MOZART chemistry provided by NCAR. The initial conditions for ice and ocean models are from CESM default settings. The initial conditions for the land model are based on the output from the NCAR CESM/CAM4 B_1850–2000_CN simulation. Table S1 in the supplementary material summarizes the model configurations including physical schemes and chemical options used in CESM-NCSU simulations. More detailed descriptions can be found in He and Zhang (2014) and

Glotfelty et al. (2017a, b). In this work, both meteorological dynamical downscaling and chemical composition downscaling from the CESM-NCSU were applied to provide meteorological and chemical ICs/BCs for regional WRF-CMAQ simulations.

Major processes for chemical composition downscaling included species mapping and horizontal and vertical interpolations. ICs were only needed for the first time step, whereas BCs were provided every 6 hours. The horizontal and vertical resolutions of CESM were 0.9 ° (latitude) × 1.25 ° (longitude) and 30 layers in hybrid sigma-pressure coordinates, respectively. Those of WRF-CMAQ were 36-km in Lambert projection coordinates and 23 layers in sigma coordinates, respectively. The horizontal interpolations to WRF-CMAQ grids were first applied by calculating distance weighted mean from four neighboring CESM grids, and the vertical interpolations to WRF-CMAQ layers were then applied by calculating pressure weighted mean from two nearest CESM layers. CESM/CAM5 and CMAQ both use the Carbon Bond 2005 (CB05) (Yarwood et al., 2005) chemical mechanism; therefore, most gas species can be directly mapped. CESM/CAM5 uses the 7-mode prognostic Modal Aerosol Model (MAM7) (Liu et al., 2012) with volatility-basis-set (VBS) (Glotfelty et al., 2017b), whereas CMAQ uses the 3-mode AERO6 aerosol module. The mapping table between CESM/CAM5 and CMAQ aerosol species is shown in Table S2. Secondary organic aerosol (SOA) species in CESM/CAM5 were divided according to different volatility levels. However, the CMAQ model includes specific SOA semi-volatile and nonvolatile species. The anthropogenic and biogenic SOA species in CESM/CAM5 were first lumped into total semi-volatile SOA and total nonvolatile SOA. The ratios among the SOA species derived from the default BCs/ICs were then used to allocate each SOA species in CMAQ based on the combined SOA, as suggested by Carlton et al. (2010). Bias-corrections were applied to the chemical ICs/BCs for species such as $O_3$ to reduce high biases against satellite retrievals (Zhang et al., 2016c). As indicated by Zhang et al. (2016c), using satellite-constrained boundary conditions for $O_3$ showed substantial improvement in model performance of tropospheric ozone residual (TOR, or column $O_3$). In this study, the boundary conditions for $O_3$ were constrained with satellite observations following the similar work of Zhang et al. (2016c). Scale factors of 0.8 to 0.95 were applied to adjust the original $O_3$ boundary conditions derived from CESM-NCSU. CESM-NCSU tends to overpredict natural dust emissions over East Asia, and modelled dust concentrations from CESM-NCSU were thus divided by 3 to reduce the high biases in dust simulations (see the Supplement).

Because GCMs generally suffer from systematic biases to a certain extent, to improve the meteorological downscaling results, meteorological ICs/BCs derived from CESM-NCSU were bias-corrected using the method developed by Yahya et al. (2016) following the work of Xu and Yang (2012), Done et al. (2015), and Bruyère et al. (2014) based on the NCEP-FNL dataset. Monthly varying mean climatological biases in ICs/BCs between CESM-NCSU and NCEP-FNL were calculated and then subtracted from the original CESM-NCSU ICs/BCs to generate bias-corrected meteorological ICs/BCs. Major variables corrected in this study included air temperature, relative humidity, zonal wind, meridional wind, geopotential height, and soil moisture because Bruyère et al. (2014) found that correcting all boundary data provides the greatest improvement. The bias-correction method assumed that the biases remain the same in the future and allowed the retention of the CESM-NUSU simulated climatic changes in the mean seasonal state, diurnal cycle, and variance of inter-annual

variation. The bias-correction method corrected the major biases in the meteorological variables that can cause serious issues for regional climate downscaling while retaining climate variability within the GCM for both current and future simulations.

## 2.3 Emissions

The CESM-NCSU simulations were driven with the RCPs emissions for both current and future decades (Glotfelty et al., 2017a). In this study, RCP 4.5 (Thomson et al., 2011) was selected as a representative scenario because it is a relatively medium scenario and aggressive emission reductions of major air pollutants in this scenario might be more suitable for China's future air quality control needs. The RCP dataset v2.0 provides global emission projections for CO (carbon monoxide), $CH_4$ (methane), $SO_2$ (sulfur dioxide), $NO_x$ (nitrogen oxides), NMVOC (non-methane volatile organic compounds), $NH_3$ (ammonia), BC (black carbon) and OC (organic carbon) as monthly averages at a spatial resolution of 0.5° for the period 2005 to 2100 with identical base year 2000 (http://tntcat.iiasa.ac.at:8787/RcpDb/dsd?Action=htmlpage&); emission sources include energy, industry, solvent use, transport, domestic combustion, agriculture, open burning of agricultural waste, waste treatment, biomass burning, shipping, and aviation sectors. In the RCP 4.5 emissions, only biomass burning, shipping, and aviation emission change monthly, and emission altitudes are given only for aviation. In order to achieve better performance for the regional WRF-CMAQ simulation, the MIX Asian emission inventory (a mosaic Asian anthropogenic emission inventory for the Model Inter-Comparison Study for Asia (MICS-Asia) and the (Hemispheric Transport of Air Pollution) HTAP projects, http://www.meicmodel.org; Li et al., 2017) for 2008 was used for the current year period (2006-2010), which has better spatial-temporal allocation profiles and particulate matter (PM) and VOC speciation profiles to generate model-ready emissions for regional scale air quality modeling over East Asia. The MIX inventory provides better monthly profiles compared to RCP 4.5 emissions and finer gridded emissions at a spatial resolution of 0.25°, which is close to the resolution of 36 km used in WRF-CMAQ. Emissions of biomass burning, shipping and aviation sector were directly used from the RCP 4.5 emissions as they were not included in the MIX inventory.

Emissions from natural sources, including biogenic VOCs emissions, soil and lightning $NO_x$ emissions, and dust emissions, were calculated inline within the two-way coupled WRF-CMAQ. The windblown dust emission scheme used in the CMAQ was developed by Tong et al. (2017). For biogenic emissions over East Asia, the Biogenic Emissions Inventory System (BEIS3) version 3.14 (Vukovich and Pierce, 2002; Schwede et al., 2005) was used in the coupled system rather than the widely used Model of Emissions of Gases and Aerosols from Nature version 2 (MEGAN2) (Guenther et al., 2012) because MEGAN2 has not been integrated into the CMAQ model. Soil $NO_x$ emissions were also calculated by the inline BEIS3 module. Lightning $NO_x$ emissions were inline calculated by estimating the number of lightning flashes based on the simulated convective precipitation (Allen et al., 2012).

**2.4 Evaluation protocols**

The model performance was evaluated against surface observations and satellite observations. Surface observations included hourly meteorological data from the National Climate Data Center (NCDC) and the real-time (i.e., hourly) concentrations of air pollutants from the China National Environmental Monitoring Center (CNEMC). The nationwide routine monitoring of

$PM_{2.5}$ in China was not initiated until 2013; CNEMC began to release hourly concentrations of CO, $SO_2$, $NO_2$, $O_3$, $PM_{2.5}$, and $PM_{10}$ in 74 major cities in China since January 2013 (http://www.cnemc.cn/), which is a much better dataset for air quality evaluation than the daily Air Pollution Index (API) dataset used in previous studies (Zhao et al., 2013; Zhang et al., 2016a). Satellite observations included data from the Global Precipitation Climatology Project (GPCP), the Clouds and the Earth's Radiant Energy System (CERES), the Moderate Resolution Imaging Spectroradiometer (MODIS), the Measurements

of Pollution in the Troposphere (MOPITT), the Ozone Monitoring Instrument (OMI), and the SCanning Imaging Absorption SpectroMeter for Atmospheric ChartographY (SCIAMACHY). The variables that were evaluated in this study included temperature at 2 m (T2), relative humidity at 2 m (RH2), wind speed at 10 m (WS10), wind direction at 10 m (WDR10), precipitation (Precip), downward shortwave radiation at the surface (SWDOWN), downward longwave radiation at the surface (LWDOWN), net shortwave radiation (GSW), outgoing longwave radiation at the top of the atmosphere (OLR),

shortwave cloud forcing (SWCF), longwave cloud forcing (LWCF), cloud fraction (CF); gas-phase species (CO, $SO_2$, $NO_2$, $O_3$), $PM_{2.5}$, $PM_{10}$; column CO, $NO_2$, $SO_2$, and HCHO; tropospheric ozone residual (TOR), and aerosol optical depth (AOD).

Two types of model evaluations were conducted in this study: evaluation for the climatological application to assess the model performance on a climatological average timescale over a 5-yr period (2006-2010) and evaluation for the short-term air quality application (2013) to assess the model performance on a monthly time scale. The observational data and

simulated data were paired on an hourly basis for air quality evaluation in 2013, whereas they were paired on a 5-year average monthly basis for climatological-type evaluation when conducting statistical analyses. Moreover, evaluation for the air quality application in 2013 focused more on surface chemical variables because more observational data were available. For the climatological application, only satellite observations of column abundance were used to assess the chemical prediction because of the shortage in surface air quality observations during 2006-2010. The performance statistical analyses

were performed following Zhang et al. (2006, 2009a, b). The statistical parameters included correlation coefficient (R), mean bias (MB), normalized mean biases (NMB), mean absolute gross error (MAGE), and root mean square error (RMSE). The statistical evaluation was in general performed for the entire regional domain. However, evaluation for surface chemical variables focused more on China where hourly air quality monitoring data are available.

**3 Results and discussion**

**3.1 Model performance for climatological application (2006-2010)**

Table 3 summarizes the performance statistics for the climatological application during the period 2006-2010. The model performed well for T2 and RH2, with MBs of -0.6 ℃ and 0.8%, correlation coefficients of 0.97 and 0.72, MAGEs of 2.4 ℃ and 9.7%, and RMSEs of 3.2 ℃ and 12.6%, respectively. From the evaluation results from Xu and Xu (2012), the Coupled Model Inter-comparison Project Phase 5 (CMIP5) multi-models tended to underpredict T2 over China with MBs ranging from –1.0 ℃ to –2.0 ℃ for the period 1961–2005. The improved statistical performance for T2 in this study compared with CMIP5 models may be related to the use of the regional model WRF and the bias-correction technique applied for CESM downscaling. This indicates that WRF-CMAQ driven by bias-corrected CESM-NCSU ICs/BCs performs well on a climatological average timescale. WS10 was moderately overpredicted by 22.2%, with an MB of 0.6 m/s, an MAGE of 1.2 m/s and a RMSE of 1.6 m/s. Large overpredictions in WS10 with NMBs of 48.7%-101.0% from WRF simulations have been reported in the literature (Penrod et al., 2014; Cai et al., 2016; Zhang et al., 2016a) because of unresolved subgrid-scale topographic features and uncertainties in parameterizations of turbulent fluxes in WRF (Hanna and Yang, 2001; Rontu, 2006; Mass and Ovens, 2011). The overpredictions in WS10 are likely caused by low surface drag due to the inappropriate representation of surface roughness because the detailed surface structure cannot be reproduced at a coarse grid resolution of 36-km. The high wind biases were reduced in this study because of the use of the simple wind correction method of Mass and Ovens (2010). The USGS 24-category land use data is out of date for China where urbanization has been dramatic, which would also partly contribute to the overprediction in WS10. Precipitation was well-predicted against GPCP with an NMB of -0.9% and moderately overpredicted by 27.4% against NCDC, and the model could generally capture the observed spatial distribution (see Figure 2). The convective precipitation dominated the overprediction of total precipitation in the southern oceanic area, which may be possibly due to overprediction of convective precipitation intensity by the Kain–Fritsch cumulus scheme. The non-convective precipitation dominated the overprediction of total precipitation in the northeastern oceanic area, which could be attributed to possible errors in the Morrison double-moment microphysics scheme. Emery et al. (2001) suggested the benchmarks for satisfactory performance for T2 (MB within ±0.5 °C, MAGE of ≤ 2.0 °C) and WS10 (MB within ± 0.5 m/s, MAGE and RMSE of 2.0 m/s). In the climatological application, the MB and MAGE of T2 and the MB of WS10 are close to the benchmark, the MAGE and RMSE of WS10 are within the benchmark, and hence the performance is deemed acceptable.

As shown in Figure 3, MBs for 5-year average T2, RH2, WS10 and precipitation were generally small over eastern China. Relatively large biases were found over the coast, especially in Japan and North and South Korea, similar to previous WRF simulations (Chen et al., 2015; Zhang et al., 2016a), which indicates certain limitations in the WRF model over complex terrain and air-sea interactions. Figure 4 compares the 5-year average monthly simulated T2, RH2, WS10 and precipitation against observations. The model could generally capture the monthly variations of T2, although there were cold biases of –1.0 ℃ in spring and summer months. The model also well reproduced the observed monthly variations of RH2; the minimum RH2 was observed in April (~ 63%) and the maximum RH2 was observed in July (~ 76%). For WS10, the model predicted a minimum in summer, which was similar to observations. However, the overprediction of WS10 in winter (with

an MB of 0.9 m/s) was slightly higher than in other seasons (with an MB of 0.5 m/s). The model accurately predicted the observed precipitation maximum occurring in summer (~ 5 mm/day) and the minimum in winter (~ 1 mm/day). Overall, the climate predictions of WRF-CMAQ represent a good approximation of the current atmosphere in terms of spatial distributions and seasonal variations, and can thus provide acceptable meteorological fields for air quality simulations.

Figure 5 compares the 5-year averaged simulated spatial distributions of cloud, aerosol and radiation variables against satellite observations. CF was underpredicted with an NMB of -30.5%, similar to previous WRF simulations over East Asia (Liu et al., 2016). Large underpredictions of CF were found over the oceanic area in the southern part of the domain. Such underpredictions may be because of the model's limitations in simulating cloud microphysics and the lack of aerosol-cloud interactions. LWDOWN was predicted very well with an NMB of -2.8%, whereas SWDOWN was overpredicted with an

NMB of 14.1%. The overpredictions of SWDOWN were likely because of underpredictions of cloud radiative forcing resulted from underpredictions of CF as well as underpredictions of aerosol direct radiative forcing resulted from underpredictions of AOD. WRF version 3.4 has neglected sub-grid cloud feedbacks to radiation, which could contribute in part to the overpredictions in SWDOWN (Alapaty et al., 2012). Overpredictions in SWDOWN generally corresponded to underpredictions in CF. Fewer clouds led to underpredictions in SWCF (with an MB of 13.6 W m$^{-2}$), which allowed more

SWDOWN to reach the ground. The overpredictions in OLR were associated with the underpredictions in LWCF (with an MB of -12.7 W m$^{-2}$). The model underpredicted AOD with an NMB of -36.3%; however, it could capture the high value over eastern China. Similar underpredictions of AOD were found over North America using offline-coupled WRF and CMAQ (Wang et al., 2012; Penrod et al., 2014) and the two-way coupled WRF-CMAQ (Hogrefe et al., 2015).

Figure 6 compares the 5-year averaged simulated spatial distributions of column mass abundance of chemical variables

against satellite retrievals. In general, the model could reproduce the spatial distributions of column mass abundance of chemical variables; correlation coefficients were generally higher than 0.8. Column CO, $NO_2$, HCHO, and $O_3$ were well predicted in terms of domain mean performance statistics, with NMBs of -11.7%, 18.3%, -4.0%, and 16.4%, respectively. Large underpredictions in column CO occurred over eastern China as well as North and South Korea and Japan, likely because of uncertainties in anthropogenic emissions as well as biomass burning (Streets et al., 2003). Simulated TOR could

capture the observed low values in the south and in Tibet and the high values in the north. The overprediction of TOR in the north can be attributed to uncertainties in upper layer BCs of $O_3$ which dominate $O_3$ concentrations in upper troposphere as well as total column $O_3$ (Zhang et al., 2016a, c). Column $NO_2$ was moderately overpredicted by 18.3%. Potential uncertainties in $NO_x$ emissions and the model treatment of deposition and chemistry processes may contribute to the model-observation difference. As discussed by Lin et al. (2010) and Han et al. (2015), there are several uncertainties in the modeled

$NO_x$ lifetime. Uncertainties in the $NO_2$ column retrievals from OMI (with a relative error of 25%, Boersma et al., 2011) and the averaging kernels (Han et al., 2015) could also help to explain the bias. Although column $SO_2$ was slightly overpredicted, with an NMB of 7.5% over the entire domain, larger overprediction occurred over eastern China. The overall error annual mean $SO_2$ columns retrieved from satellites could be as large as 45%–80% in polluted regions (Lee et al., 2009), which

might impact the evaluation results of column $SO_2$. Large uncertainties in $SO_2$ emissions (Hong et al., 2017) would also contribute to the biases in column $SO_2$. Column HCHO over eastern China was well predicted in terms of both the magnitude and the spatial pattern. Possible reasons for the overpredictions of column HCHO in Southeast Asia and northeastern India include uncertainties in biogenic and anthropogenic VOCs emissions, and satellite retrievals.

## 3.2 Model performance for short-term air quality application in 2013

Table S3 summarizes performance statistics of meteorological variables for the air quality application in January, April, and July, 2013. The model performance for major meteorological variables in the air quality application was similar to that for the climatological application. Note that for the air quality application driven with NCEP-FNL data, the observation and simulation data pairs for surface meteorological variables against NCDC observational data were on an hourly basis. The high correlations for major meteorological variables in Table S3 indicated that the model showed good skills in hourly meteorological predictions, thus NCEP-FNL data were sufficient to support air quality applications for hourly air quality predictions.

Table 4 summarizes performance statistics of chemical variables for the air quality application in January, April and July, 2013. The model performed very well for surface concentrations of $SO_2$, $NO_2$, $PM_{2.5}$ in January and April, with NMBs generally within ±10%, and moderately overpredicted $O_3$ and $PM_{2.5}$ in July. Surface CO concentrations were underpredicted in all months with NMBs ranging from -48% to -33%. The simulated column abundances of CO were also underpredicted with NMBs ranging from -14% to -2%, indicating that CO emissions were likely underestimated. Overpredictions in WS10 shown in Table S3 also contributed to the underpredictions in surface CO concentrations. Surface $SO_2$ and $NO_2$ concentrations were largely overpredicted in summer with NMBs higher than 40%, especially at nighttime, which could be attributed to biases in meteorological predictions (e.g., turbulent mixing) (Pleim, et al., 2015), uncertainties in emissions (e.g., monthly profiles and vertical distributions) (Zhang et al., 2016a) and biases in model treatments (e.g., $SO_2$ wet deposition). $PM_{2.5}$ concentrations were slightly overpredicted in winter with an NMB of 6.4%. $O_3$ was moderately overpredicted in summer with an NMB of 18.2%, which could be partly because of overpredictions in SWDOWN and partly because of overpredicted $NO_2$. Figure 7 shows the spatial distribution of $PM_{2.5}$ and $O_3$ in January, April, and July, 2013. High $PM_{2.5}$ concentrations were predicted in winter over the North China Plain (NCP) and the Sichuan Basin (SCB) where anthropogenic emissions are high, consistent with the observational data. $PM_{2.5}$ concentrations over western China were predicted to be higher in April because of spring dust emissions that could contribute not only coarse PM but also fine PM. In summer, $O_3$ concentrations were predicted to be higher over northern China but lower over southern China because the East Asian summer monsoon (EASM) brings clean air masses from the oceans in the south (He et al., 2008; Wang et al., 2011). The simulated spatial distribution of $O_3$ was consistent with observational data. Figure 8 shows the time series of hourly concentrations of $PM_{2.5}$ in January and $O_3$ in July over three heavily-polluted regions in China: the Beijing-Tianjin-Hebei (BTH) region, the Yangtze River Delta (YRD) and the Pearl River Delta (PRD). The model well reproduced the

observed hourly variations of PM$_{2.5}$ as well as the observed diurnal and daily variations of O$_3$ over three key regions. The simulated diurnal variability of PM$_{2.5}$ in BTH and YRD is somewhat larger than observations. The overprediction in surface PM$_{2.5}$ concentration at nighttime might be partly attributed to errors in the parameterization of PBL turbulent mixing (Pleim, et al., 2015). The discrepancies between the simulated and observed time series of PM$_{2.5}$ may be attributable to several possible causes, including inaccuracies in meteorological predictions (e.g., turbulent mixing, precipitation, and WS10) and uncertainties in some model treatments (e.g., secondary organic aerosol formation and dry and wet deposition). The model slightly overpredicted the peak concentrations of O$_3$, which may be partly because of overpredictions in SWDOWN.

The CMAQ performance of chemical predictions in this study was comparable to or even better than those of other air quality studies over East Asia (Wang et al., 2009; 2012; Liu et al., 2010; Zheng et al., 2015; Hu et al., 2016; Liu et al., 2016; Zhang et al., 2016a). This study predicted relatively well for most chemical species in most months. Compared with other regional modeling studies, WRF-CMAQv5.0.2 used in this study outperformed MM5/CMAQv4.6, which tend to underpredict the surface concentrations of major species with NMBs generally greater than -40% and overpredict surface O$_3$ concentrations in most months with NMBs generally higher than 20% over East Asia according to the evaluation results of Zhang et al. (2016a). A relatively good performance of CMAQv5.0.1 was also reported by Hu et al. (2016). Global models such as GEOS-Chem and CESM tend to underpredict PM$_{2.5}$ concentrations (by about 50% as reported by Jiang et al., 2013) and overpredict O$_3$ concentrations (by about 50% as reported by He and Zhang, 2014; Wang et al., 2013) in China/East Asia because of relatively coarse grid resolution and limitations in some model treatments (e.g., missing emissions of unspeciated primary PM$_{2.5}$, and discrepancies in surface layer height and vertical mixing). The comparison of the model performances for PM$_{2.5}$ and O$_3$ predictions of WRF-CMAQ and CESM is shown in Figure 9. Compared with global models, WRF-CMAQ was able to capture the high PM$_{2.5}$ concentrations in urban areas where most observational data were obtained. The model was also able to predict low O$_3$ concentrations and predicted well for O$_3$ over China with small NMBs of 15-30% in both winter and summer. CESM tended to underpredict PM$_{2.5}$ concentrations over China with NMBs ranging from -30% to -70% and overpredict O$_3$ concentrations with NMBs ranging from 50% to 100%. It should be noted that although the years of observational data and CESM simulations were not consistent (i.e., 2013 and 2006-2010, respectively), we do not think inter-annual changes in meteorological fields and emissions contributed to such large biases, as is indicated by the results of the two WRF-CMAQ simulations for the two periods (see Figure 9).

### 3.3 Improvements of model performance within the modeling system

To demonstrate the model improvements made in this study, sensitivity simulations were conducted. The comparison of the baseline simulation (i.e., NCEP_BASE) and the sensitivity simulation (i.e., NCEP_BASE_WoImp) against observational data is shown in Figure 10. The coupling model system predicted AOD relatively well. The two-way coupled WRF-CMAQ with aerosol direct feedbacks could generally replicate lower SWDOWN values over heavily-polluted regions (such as eastern and southern China). The overprediction of SWDOWN in January in the sensitivity simulation without aerosol

feedbacks (with an NMB of 19.9%) was significantly reduced when the aerosol feedbacks were included (with an NMB of 11.1%). The remaining overprediction in SWDOWN in the NCEP_BASE simulation was because of underpredictions in AOD and CF, which indicates that including the aerosol feedback in the coupled system is important for better simulating the shortwave radiation fields in WRF, consistent with the findings of Yahya et al. (2016).

The chemical composition downscaling approach was applied in this study to provide dynamical chemical BCs for regional modeling. The main advantage of applying chemical composition downscaling is the representation of global changes in atmospheric composition in regional simulations, which is important to better simulate relatively long-lived species such as CO and $O_3$ under a globally changing atmospheric environment. Another advantage is the representation of spatial and temporal variations in BCs, which could also help improve the model performance (Tang et al., 2009). The comparison

between BCs derived from CESM and fixed boundary profiles provided by the operational CMAQ system is shown in Fig. S2. CESM-derived BCs produced better spatial variability, such as higher $O_3$ concentrations from the northern boundary and lower $O_3$ concentrations from the southern boundary. When using BCs derived from CESM, the model performance of column variables (e.g., TOR) was improved in terms of spatial distribution and seasonal variations. The overprediction of TOR in January in the sensitivity simulation using fixed BCs (with an NMB of 48.9%) was significantly reduced when using

CESM-derived BCs (with an NMB of 22.3%).

Inline emissions from natural sources were calculated within the coupled system. Although BEIS3 has been widely used in the U.S, the model performance over other regions such as East Asia should be evaluated. We conducted the sensitivity simulation using offline MEGAN2, which has been widely used over East Asia. The major differences in emissions were found for isoprene emissions (see Figure 11). The summertime isoprene emissions over China estimated using MEGAN2

were approximately 100% higher than those estimated using BEIS3. Similar large discrepancies in isoprene emissions were also found from previous studies over the U.S. (Lam et al., 2011; Hogrefe et al., 2011) because of the different methods used to estimate isoprene emissions (Lam et al., 2011). We evaluated CMAQ-simulated HCHO columns using the BEIS3 and MEGAN2 emissions against OMI satellite observations. HCHO columns have been used to evaluate biogenic VOC emission inventories (Han et al., 2013) because HCHO is an intermediate oxidation product of anthropogenic and biogenic VOCs. The

evaluation results in Figure 10 show that using BEIS3 emissions in CMAQ could capture both the magnitude and the spatial pattern of HCHO columns from OMI, whereas using MEGAN2 emissions resulted in 30%~50% overpredictions of HCHO columns over northern China.

The default dust scheme in CMAQ developed by Tong et al. (2017) underpredicted dust emissions over East Asia (Fu et al., 2014; Dong et al., 2015). In this study, the dust module was revised with updated friction velocity thresholds to avoid double

counting of the impacts of soil moisture (Dong et al., 2015). Compared with sensitivity simulation results using the default dust scheme, the revised model was able to simulate springtime dust emissions over northwest China, where dust storms often occur. As shown in Figure 10, the revised model predicted AOD values in April as high as 0.2~0.6 over northwest China, which were much closer to the satellite observations, while the original model generally predicted AOD values less

than 0.05 over northwest China. The improved dust module was able to capture the spatial distribution and the temporal variations of dust emissions, although some biases still existed.

## 3.4 Aerosol direct effects on regional climate and air quality

To examine the aerosol direct effects on regional climate and air quality, we conducted a sensitivity simulation without aerosol feedback. The differences between the simulations with and without aerosol direct feedback (i.e., CESM_BASE: with feedback and CESM_BASE_Sens: without feedback) are shown in Figure 12. Aerosol direct radiative effects resulted in a reduction of shortwave radiation reaching the surface because of aerosol extinction (i.e., scattering and absorbing). The aerosol extinction led to a more stable planetary boundary layer (PBL) during the haze episode through enhancing the temperature inversion in two ways: diminished surface solar radiation led to a decrease of air temperature at the surface, and the absorption of light-absorbing particles such as black carbon (BC) caused an increase of air temperature in the upper PBL. As shown in Figure 12, the domain mean reductions in SWDOWN were -7.5 W m$^{-2}$ in January and -7.0 W m$^{-2}$ in July. The domain mean reductions in T2 were -0.09 ℃ in January and -0.08 ℃ in July. The effects of anthropogenic aerosols on SWDOWN and T2 were comparable to the results over East Asia from WRF/Chem-MADRID (Liu et al., 2016) and WRF/Chem (Zhang et al., 2016b). The reductions in SWDOWN in July were somewhat smaller than those of Liu et al. (2016) because aerosol indirect effects were not considered in this study. We also found that SWDOWN decreased in July in northwest China because of the natural dust aerosols. Slight increases in SWDOWN and T2 occurred in July in some areas, which could be attributed to semi-indirect effects of aerosols (Forkel et al., 2012). The aerosol feedbacks were significant over heavily-polluted regions such as eastern China and the Sichuan Basin. With the aerosol feedbacks, the monthly mean SWDOWN and planetary boundary layer height (PBLH) in January decreased by 21.8 W m$^{-2}$ (14%) and 35.7 m (7.6%), respectively, in major cities of China, and air temperature at the surface decreased by 0.45 ℃.

The aerosol direct feedbacks affect not only climate but also air quality because of changing climate. We investigated the aerosol direct effects on air quality in different seasons. Enhanced PBL stability resulted from the aerosol direct effects enhanced the air pollution by suppressing the dispersion of air pollutants. Because of aerosol feedbacks, mean concentrations of major pollutants (except for $O_3$) over major cities of China increased by 4.8%~9.5%, and $PM_{2.5}$ concentrations increased by 6.6 μg m$^{-3}$ in January. However, $O_3$ concentrations in January decreased by 5.1% because of aerosol feedbacks, which may be attributed to the increased $NO_x$ titration resulted from increased atmospheric stability and reduced PBL height. Similar aerosol direct effects were also found in July. Because of aerosol feedbacks, mean concentrations of major pollutants (except for $O_3$) increased by 4.8%~7.1% over major cities in China, and $PM_{2.5}$ concentrations increased by 3.8 μg m$^{-3}$ in July. The aerosol direct effects on $PM_{2.5}$ concentrations in July were smaller than those in January because of lower aerosol loadings in July. Compared with simulated aerosol effects over the continental U.S. and Europe (Zhang et al., 2010; Hogrefe et al., 2015), the magnitudes of aerosol effects on regional climate and air quality were much larger over East Asia because of higher aerosol loadings resulted from severe regional pollution.

## 4 Conclusions

A regional coupled climate-chemistry modeling system using the dynamical downscaling technique was established by linking the global CESM model and the regional two-way coupled WRF-CMAQ model for the purpose of comprehensive assessments of regional climate change and air quality and their interactions within one modeling framework. The modeling
system took full advantage of global climate-chemistry models that can well predict large-scale global changes and regional models that can better represent regional phenomena. The modeling system was applied over East Asia for a multiyear climatological application during 2006-2010 under RCP 4.5 as well as a short-term air quality application for three months in 2013 driven by the NCEP-FNL reanalysis dataset. Comprehensive model evaluation was conducted against surface observations and satellite observations to assess the model's performance.

The two-way coupled WRF-CMAQ generally performed well for both the climatological and the short-term air quality applications. The model was able to predict major meteorological variables satisfactorily. The improved statistical performance for T2 in this study (with an MB of -0.6 ℃) compared with CMIP5 multi-models may be related to the use of the regional model WRF and the bias-correction technique applied for CESM downscaling. The model showed good ability to predict $PM_{2.5}$ in winter (with an NMB of 6.4% in 2013) and $O_3$ in summer (with an NMB of 18.2% in 2013) in terms of
statistical performance and spatial distributions. Compared with global models that tend to underpredict $PM_{2.5}$ concentrations in China, WRF-CMAQ was able to capture the high $PM_{2.5}$ concentrations in urban areas. Model improvements made in this study were quantified by the sensitivity simulation. The coupled modeling system with direct aerosol feedbacks predicted AOD relatively well and significantly reduced the overprediction of SWDOWN (NMBs in January were reduced from 19.9% to 11.1%). The two-way coupled WRF-CMAQ with aerosol direct feedbacks could generally replicate lower SWDOWN
values over heavily polluted regions (such as eastern and southern China). Applying chemical composition downscaling to introduce global background changes in atmospheric composition could also help improve the model performance of column variables (e.g., TOR). The overprediction of TOR in January when using fixed BCs (with an NMB of 48.9%) was significantly reduced when using CESM-derived BCs (with an NMB of 22.3%). The BEIS3 biogenic online emission module was applied in this study, and the model performance over East Asia was examined. Sensitivity simulations showed
that using BEIS biogenic emissions resulted in improved performance for column HCHO, whereas using MEGAN2 emissions resulted in large overpredictions (30%~50%) of HCHO columns over northern China. The improved dust module was able to capture the spatial distribution and the temporal variations of dust emissions, although some biases remained. The revised model was able to capture the high AOD values (0.2~0.6) in April over northwest China where dust storms often occur in spring. We also demonstrated the impacts of aerosol direct effects on climate and air quality to address important
climate-chemistry interactions. With aerosol direct feedbacks in January, the monthly mean SWDOWN and PBLH over major cities of China decreased by 21.8 W m$^{-2}$ (14%) and 35.7 m (7.6%), respectively, air temperature at the surface decreased by 0.45 ℃, and mean concentrations of most pollutants (except for $O_3$) increased by 4.8%-9.5%. The aerosol effects on climate and air quality were more significant in East Asia than the U.S. and Europe because of higher aerosol

loadings resulting from severe pollution in East Asia, which indicates the need to apply online-coupled models over East Asia for regional climate and air quality modeling and to study the important climate-chemistry interactions.

This work has established the baseline simulation for WRF-CMAQ application for a future time period in order to access the projected changes in climate and anthropogenic emissions on future air quality over East Asia under the RCP4.5 scenario. Although the modeling system generally had acceptable performance, this work suggested further model development and improvement that could improve the model performance. First, this work used a highly simplified method to correct wind bias, a more rigorous method that is available in WRF version 3.6 should be used. Second, larger biases were found for cloud fraction against satellite data and also for surface $SO_2$ concentrations during summer against surface observations. The performance of cloud variables was not as good as that of other meteorological variables, and underpredictions of cloud fraction resulted in overpredictions of SWDOWN and underpredictions of shortwave and longwave cloud forcing. The model biases possibly resulted from uncertainties in simulated meteorology (e.g., precipitation and WS10), emissions (e.g., vertical profiles and biogenic emissions), boundary conditions derived from the global CESM model, and limitations in some model treatments (e.g., cumulus scheme, secondary organic aerosol). Further model improvement should focus on these areas identified from this work. Finally, aerosol indirect effects on cloud properties are currently not included in the released version of the two-way coupled WRF-CMAQ model. An initial implementation and evaluation of aerosol indirect effects on resolved clouds over the U.S. has recently been completed (Yu et al., 2014), but its performance outside the U.S. needs to be further evaluated in subsequent studies.

**Code availability**

The two-way coupled WRF-CMAQ model is open-source and publicly available. The WRF version 3.4 codes can be downloaded at http://www2.mmm.ucar.edu/wrf/users/download/get_source.html. The CMAQ version 5.0.2 codes and the WRF-CMAQ two-way package can be downloaded at https://www.cmascenter.org/download.cfm. The build instructions and run instructions for the two-way coupled WRF-CMAQ model are available at http://www.airqualitymodeling.org/cmaqwiki/index.php?title=CMAQv5.0.2_Two-way_model_release_notes. We have modified the surface drag parameterization in WRF3.4 for correction of wind speed bias and the dust module in CMAQv5.0.2 to generate more dust emissions. The modified codes can be provided upon request.

*Acknowledgements.* This study was sponsored by China's National Basic Research Program (2014CB441301), China's National Key R&D Research Program (2016YFC0208801), and the National Natural Science Foundation of China (41625020 and 41222036). The work at NCSU was supported by the National Science Foundation EaSM program (AGS-1049200). Chaopeng Hong acknowledges the support from Tsinghua Scholarship for Overseas Graduate Studies (2014203). The CESM simulations were conducted by Tim Glotfelty at NCSU. Thanks are due to Khairunnisa Yahya and Tim Glotfelty at NCSU for their help during the generation of the initial and boundary conditions from CESM for WRF-CMAQ

simulations. Thanks are due to Kai Wang, Jian He, and Patrick Campbell at NCSU for their help during the setup of the modeling system. Thanks are due to Ruby Leung at PNNL for providing the script to generate meteorological initial and boundary conditions from CESM to WRF. The authors acknowledge high-performance computing support from Yellowstone (ark:/85065/d7wd3xhc) provided by NCAR's Computational and Information Systems Laboratory, sponsored

by the National Science Foundation and Information Systems Laboratory.

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

**Table 1. Model configurations and set-up for the climatological application.**

| Model Attribute | Configuration |
| --- | --- |
| Model | Two-way Coupled WRF3.4 and CMAQv5.0.2 (Wong et al., 2012) |
| Domain and Resolutions | 36 km × 36 km over East Asia; 23 sigma layers from surface to 100 mb |
| Simulation Period | Current years (2006-2010) |
| Meteorological and Chemical ICs/BCs | The NCSU's CESM/CAM5 v1.2.2 (Gantt et al., 2014; He and Zhang, 2014; He at al., 2015a, b; Glotfelty et al., 2017a, b; Glotfelty and Zhang, 2017); meteorological ICs/BCs are bias-corrected with NCEP FNL data based on Xu and Yang (2012) |
| Anthropogenic Emissions | MIX Asian 2008 emission inventory (a mosaic Asian anthropogenic emission inventory for the MICS-Asia and the HTAP projects, http://www.meicmodel.org; Li et al., 2017) for current years |
| Biogenic Emissions | BEISv3.1.4 (Vukovich and Pierce, 2002; Schwede et al., 2005) |
| Dust Emissions | The physically based dust emission algorithm, FENGSHA (Tong et al., 2017) |
| Radiation | RRTMG shortwave (SW) and longwave (LW) (Iacono et al., 2008) |
| PBL | ACM2 PBL scheme (Pleim, 2007) |
| Land Surface | Pleim–Xiu land surface model (Xiu and Pleim, 2001) |
| Surface Layer | Pleim–Xiu surface layer scheme |
| Land Use Category | The USGS 24-category land use data |
| Microphysics | Morrison double-moment (Morrison et al., 2009) |
| Cumulus Parameterization | Kain–Fritsch cumulus scheme (Kain, 2004) |
| Gas-Phase Chemistry | CB05 gas-phase mechanism with active chlorine chemistry and updated toluene mechanism (Yarwood et al., 2005; Whitten et al., 2010) |
| Aerosol Module | AERO6 (Appel et al., 2013) |

**Table 2. Simulation design.**

| Run index | Simulated period | Model configuration | Purpose |
|---|---|---|---|
| *Climatological applications (2006-2010) with CESM ICs/BCs* | | | |
| CESM_BASE (Production runs) | 2006-2010 | See Table 1, with CESM ICs/BCs and improvements made in NCEP_BASE | Evaluate model performance for climatological applications |
| CESM_BASE_Sens | Jan., 2008; Jul., 2007 | Same as CESM_BASE but without aerosol feedback | Access the aerosol direct effects on regional climate and air quality |
| *Air quality applications in 2013 with NCEP ICs/BCs* | | | |
| NCEP_BASE | 2013 (Jan., Apr., Jul.) | With NCEP ICs/BCs and several improvements (refer to Section 2.1) | Evaluate model performance for air quality applications |
| NCEP_BASE_WoImp | 2013 (Jan., Apr., Jul.) | Without updated improvements | Compared with NCEP_BASE, to support the improvements |

**Table 3. Model performance statistics for the climatological application (2006-2010, CESM_BASE).**

| Variable | Network | MeanObs | MeanSim | R | MB | NMB (%) | MAGE | RMSE |
|---|---|---|---|---|---|---|---|---|
| T2 (°C) | NCDC | 14.2 | 13.6 | 0.97 | -0.6 | -4.6 | 2.4 | 3.2 |
| RH2 (%) | NCDC | 69.0 | 69.8 | 0.72 | 0.8 | 1.1 | 9.7 | 12.6 |
| WS10 (m s$^{-1}$) | NCDC | 2.7 | 3.3 | 0.47 | 0.6 | 22.2 | 1.2 | 1.6 |
| WDR10 (degree) | NCDC | 210.7 | 186.8 | 0.35 | -23.8 | -11.3 | 37.2 | 60.2 |
| Precip (mm day$^{-1}$) | NCDC | 2.5 | 3.2 | 0.52 | 0.7 | 27.4 | 1.4 | 2.1 |
| Precip (mm day$^{-1}$) | GPCP | 3.0 | 3.0 | 0.80 | 0.0 | -0.9 | 0.8 | 1.3 |
| SWDOWN (W m$^{-2}$) | CERES | 184.5 | 210.5 | 0.90 | 26.0 | 14.1 | 26.0 | 29.4 |
| LWDOWN (W m$^{-2}$) | CERES | 330.1 | 320.8 | 0.99 | -9.4 | -2.8 | 10.0 | 12.9 |
| GSW (W m$^{-2}$) | CERES | 157.0 | 171.9 | 0.91 | 14.8 | 9.4 | 17.3 | 20.9 |
| OLR (W m$^{-2}$) | CERES | 235.0 | 244.0 | 0.89 | 9.0 | 3.8 | 10.3 | 13.3 |
| SWCF (W m$^{-2}$) | CERES | -50.5 | -36.8 | 0.74 | 13.6 | -27.0 | 14.5 | 18.6 |
| LWCF (W m$^{-2}$) | CERES | 29.9 | 17.3 | 0.20 | -12.7 | -42.3 | 12.7 | 15.1 |
| CF (%) | MODIS | 64.7 | 45.0 | 0.11 | -19.7 | -30.5 | 20.9 | 25.6 |
| Column CO ($10^{15}$ molec. cm$^{-2}$) | MOPITT | 2075.3 | 1832.2 | 0.83 | -243.1 | -11.7 | 324.2 | 376.2 |
| Column NO$_2$ ($10^{15}$ molec. cm$^{-2}$) | OMI | 1.5 | 1.8 | 0.91 | 0.3 | 18.3 | 0.6 | 1.2 |
| Column SO$_2$ ($10^{15}$ molec. cm$^{-2}$) | SCIA | 5.5 | 5.9 | 0.82 | 0.4 | 7.5 | 3.6 | 6.1 |
| Column HCHO ($10^{15}$ molec. cm$^{-2}$) | OMI | 5.9 | 5.7 | 0.87 | -0.2 | -4.0 | 1.3 | 1.8 |
| TOR (DU) | OMI | 31.2 | 36.3 | 0.92 | 5.1 | 16.4 | 5.3 | 5.9 |
| AOD | MODIS | 0.3 | 0.2 | 0.82 | -0.1 | -36.3 | 0.1 | 0.1 |

[1] Mean Obs: Mean observed data; Mean Sim: Mean simulated data; R: correlation coefficient; MB: mean bias; NMB: normalized mean biases; MAGE: mean absolute gross error; RMSE: root mean square error.

**Table 4. Model performance statistics for the air quality application: chemical variables (2013, NCEP_BASE).**

| Variable | Network | January | | | | April | | | | July | | | |
|---|---|---|---|---|---|---|---|---|---|---|---|---|---|
| | | R | MB | NMB (%) | RMSE | R | MB | NMB (%) | RMSE | R | MB | NMB (%) | RMSE |
| CO (mg m$^{-3}$) | CNEMC | 0.5 | -0.8 | -34.9 | 1.8 | 0.2 | -0.5 | -48.1 | 1.0 | 0.2 | -0.3 | -33.1 | 0.8 |
| SO$_2$ (μg m$^{-3}$) | CNEMC | 0.3 | -0.3 | -0.4 | 102.4 | 0.2 | 2.8 | 9.2 | 40.5 | 0.1 | 16.3 | 89.1 | 47.6 |
| NO$_2$ (μg m$^{-3}$) | CNEMC | 0.4 | -1.4 | -2.0 | 44.1 | 0.4 | -0.2 | -0.5 | 35.3 | 0.3 | 12.8 | 43.6 | 37.9 |
| O$_3$ (μg m$^{-3}$) | CNEMC | N/A | N/A | N/A | N/A | 0.3 | -3.1 | -4.4 | 59.7 | 0.5 | 11.2 | 18.2 | 54.7 |
| PM$_{2.5}$ (μg m$^{-3}$) | CNEMC | 0.5 | 8.9 | 6.4 | 112.7 | 0.4 | -1.6 | -2.8 | 47.4 | 0.4 | 11.5 | 28.6 | 48.5 |
| PM$_{10}$ (μg m$^{-3}$) | CNEMC | 0.5 | -27.9 | -13.8 | 137.1 | 0.3 | -28.4 | -24.9 | 99.9 | 0.4 | -5.6 | -7.6 | 68.3 |
| Column CO ($10^{15}$ molec. cm$^{-2}$) | MOPITT | 0.8 | -304.6 | -13.7 | 537.4 | N/A | N/A | N/A | N/A | 0.3 | -30.2 | -1.9 | 540.6 |
| Column NO$_2$ ($10^{15}$ molec. cm$^{-2}$) | OMI | 0.9 | -0.4 | -13.8 | 3.4 | 0.9 | -0.1 | -3.2 | 1.6 | 0.8 | 0.2 | 16.1 | 1.4 |
| Column SO$_2$ ($10^{15}$ molec. cm$^{-2}$) | OMI | 0.7 | 2.2 | 29.2 | 13.9 | 0.6 | -1.7 | -26.8 | 5.7 | 0.6 | -3.8 | -66.4 | 5.2 |
| Column HCHO ($10^{15}$ molec. cm$^{-2}$) | OMI | 0.7 | -1.7 | -28.5 | 2.8 | 0.8 | -0.5 | -8.9 | 2.5 | 0.7 | 1.6 | 28.1 | 3.2 |
| TOR (DU) | OMI | 0.4 | 5.9 | 22.3 | 9.3 | 0.3 | 4.1 | 12.1 | 10.3 | 0.8 | 5.7 | 16.7 | 7.7 |
| AOD | MODIS | 0.6 | 0.0 | -6.4 | 0.2 | 0.5 | -0.1 | -30.3 | 0.2 | 0.5 | -0.1 | -31.5 | 0.2 |

[1] R: correlation coefficient; MB: mean bias; NMB: normalized mean biases; RMSE: root mean square error; N/A: Data not available.

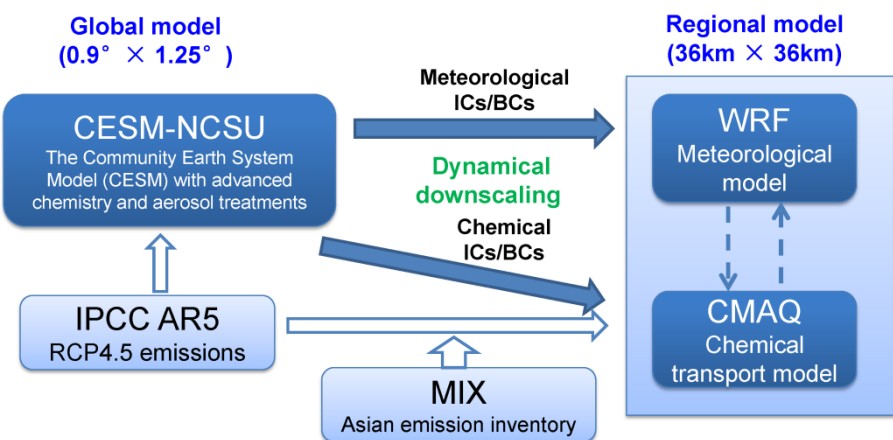

Figure 1. Modeling system in this study. The **two-way** coupled Weather Research and Forecasting - Community Multiscale Air Quality (WRF-CMAQ) model, which takes into account the air quality and climate interactions, is driven by the Community Earth System Model with advanced chemistry and aerosol treatments (CESM-NCSU) for high-resolution regional simulations. Both meteorological downscaling and chemical composition downscaling from the CESM-NCSU are applied to provide meteorological and chemical boundary conditions (BCs)/initial conditions (ICs) for regional WRF-CMAQ simulations.

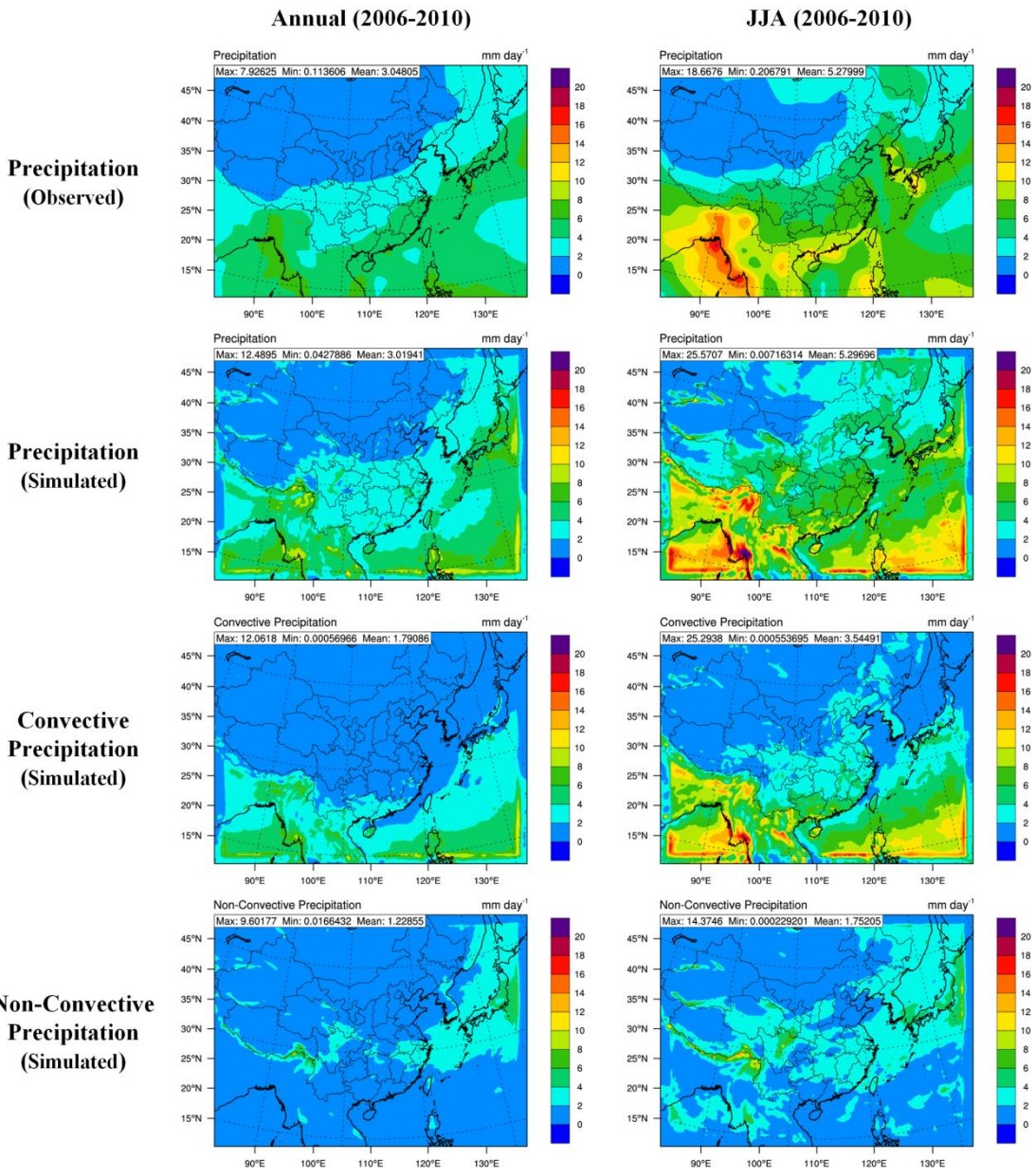

**Figure 2.** Spatial distribution of satellite-derived precipitation from GPCP, simulated precipitation, convective precipitation and non-convective precipitation under the climatological application during 2006-2010.

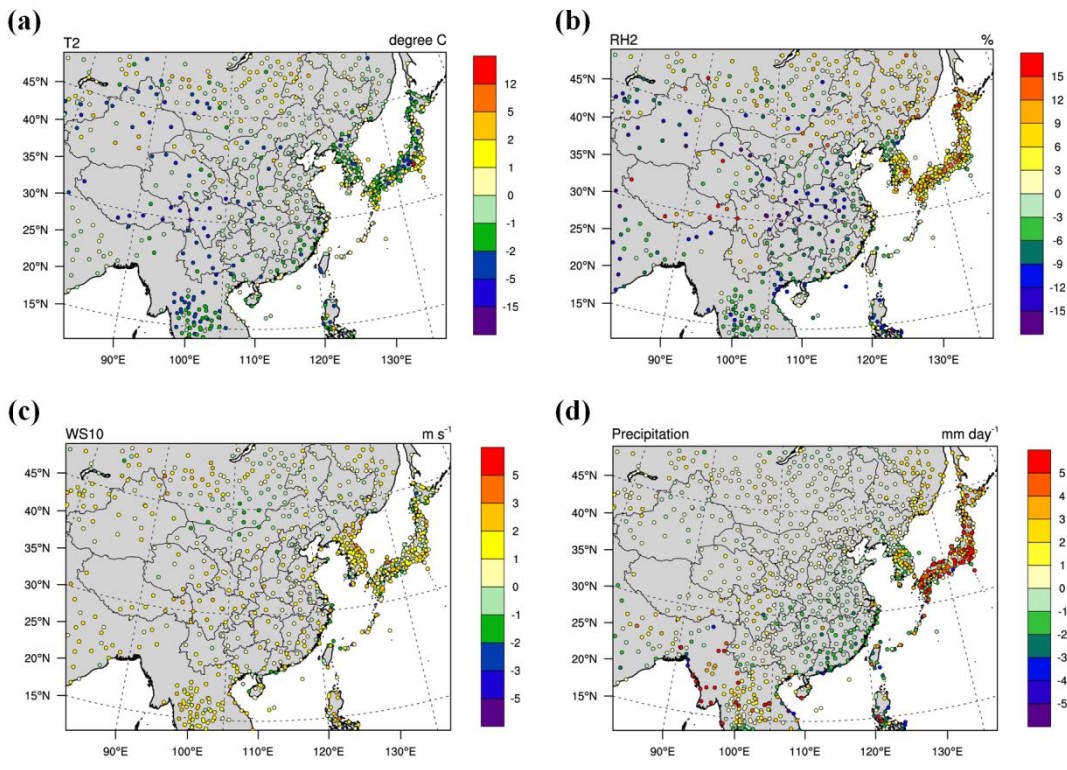

**Figure 3. Spatial distribution of MBs for (a) 2-m temperature (T2), (b) 2-m relative humidity (RH2), (c) 10-m wind speed (WS10), and (d) precipitation from NCDC under the climatological application during 2006-2010. Each marker represents the MB of each variable at each observational site.**

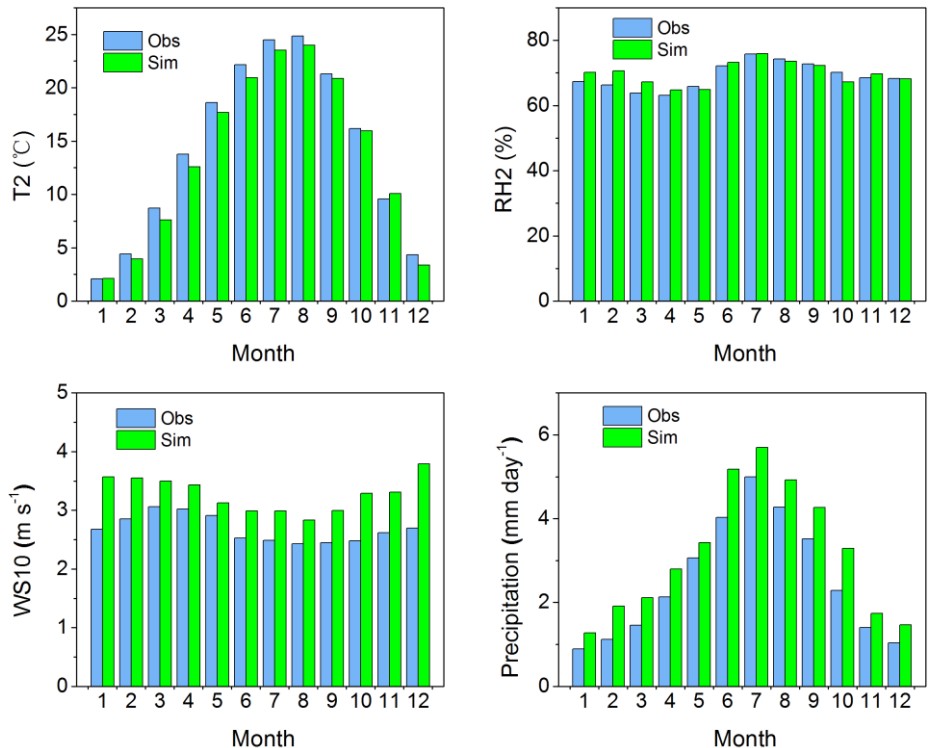

**Figure 4. 5-year averaged monthly observations (blue) vs. simulations (green) for (a) 2-m temperature (T2), (b) 2-m relative humidity (RH2), (c) 10-m wind speed (WS10), and (d) precipitation under the climatological application during 2006-2010.**

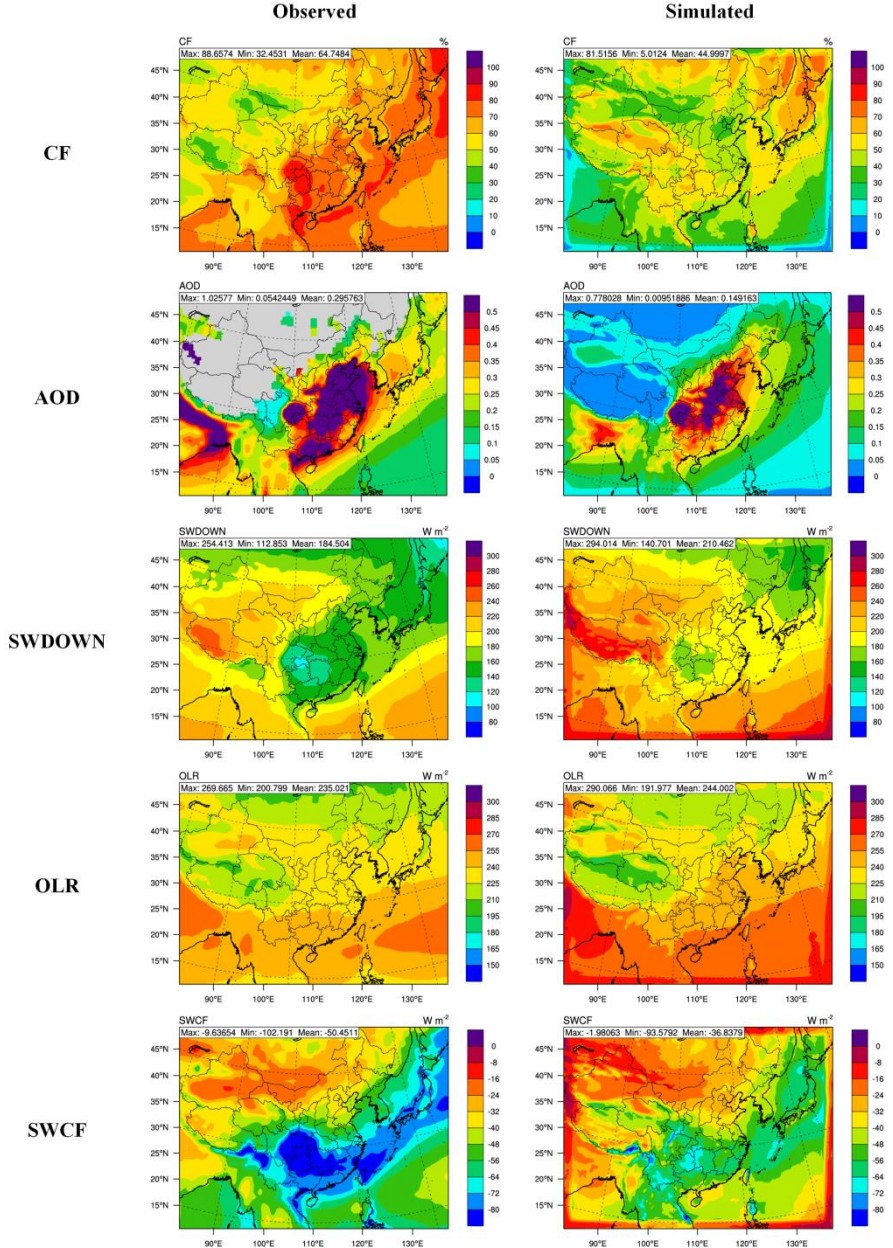

**Figure 5. Spatial distribution of satellite-derived and simulated multi-year means of CF, AOD, SWDOWN, OLR and SWCF under the climatological application for the time period 2006-2010.**

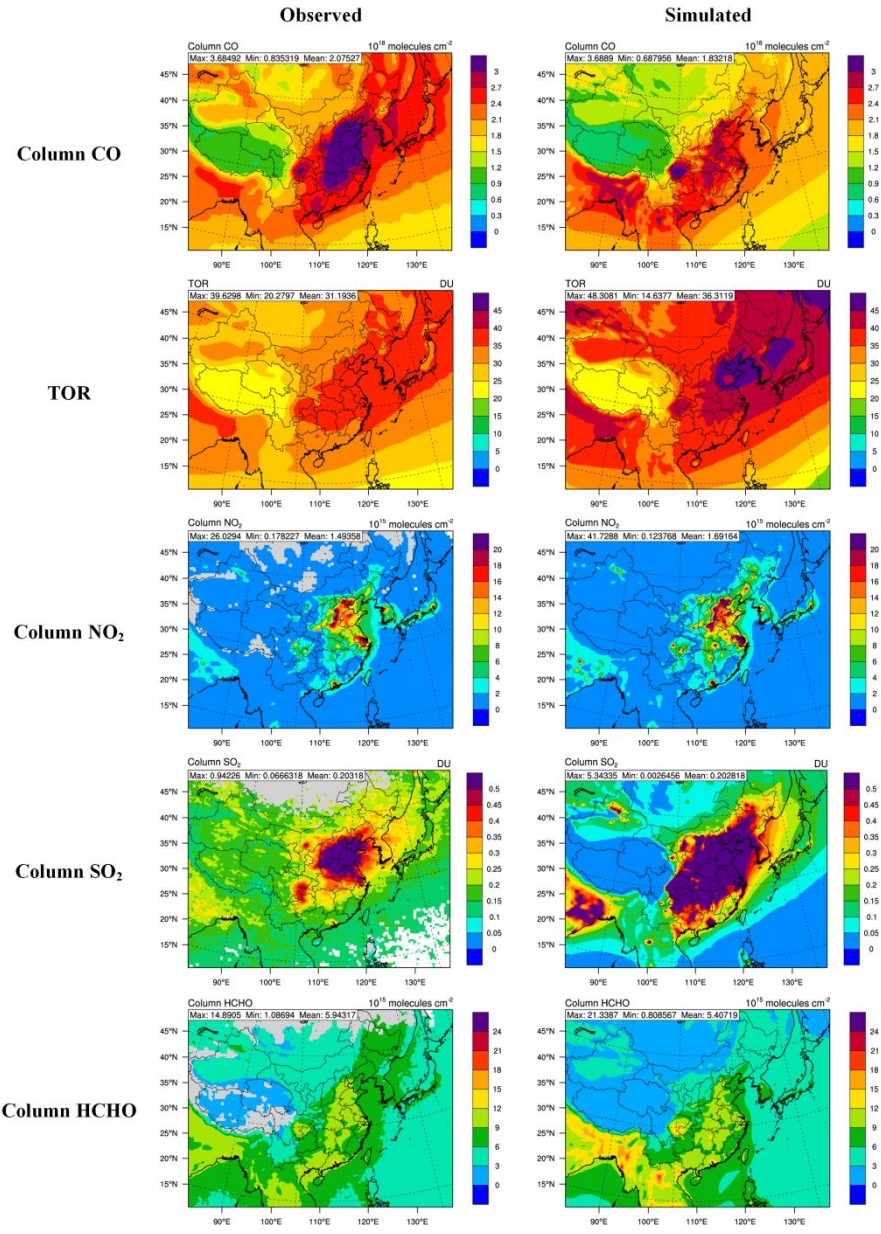

**Figure 6. Spatial distribution of satellite-derived and simulated multi-year means of column CO, TOR, column NO$_2$, column SO$_2$, and column HCHO under the climatological application for the time period 2006-2010.**

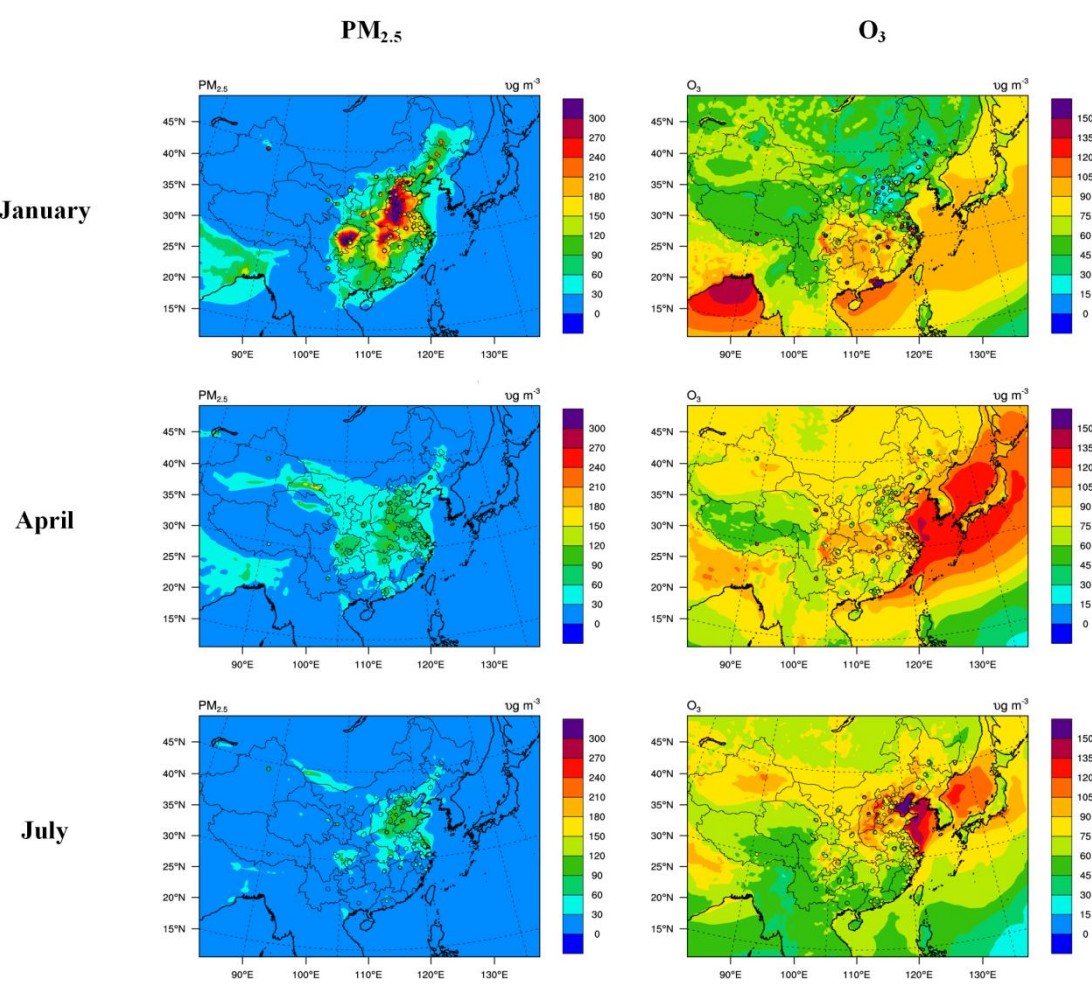

**Figure 7. Spatial distribution of observed vs. simulated PM₂.₅ and O₃ concentration during January, April and July, 2013, under the short-term air quality application. The background plots represent the simulated data, whereas observations are represented by the markers. Note that there were some errors in O₃ observational data in January 2013.**

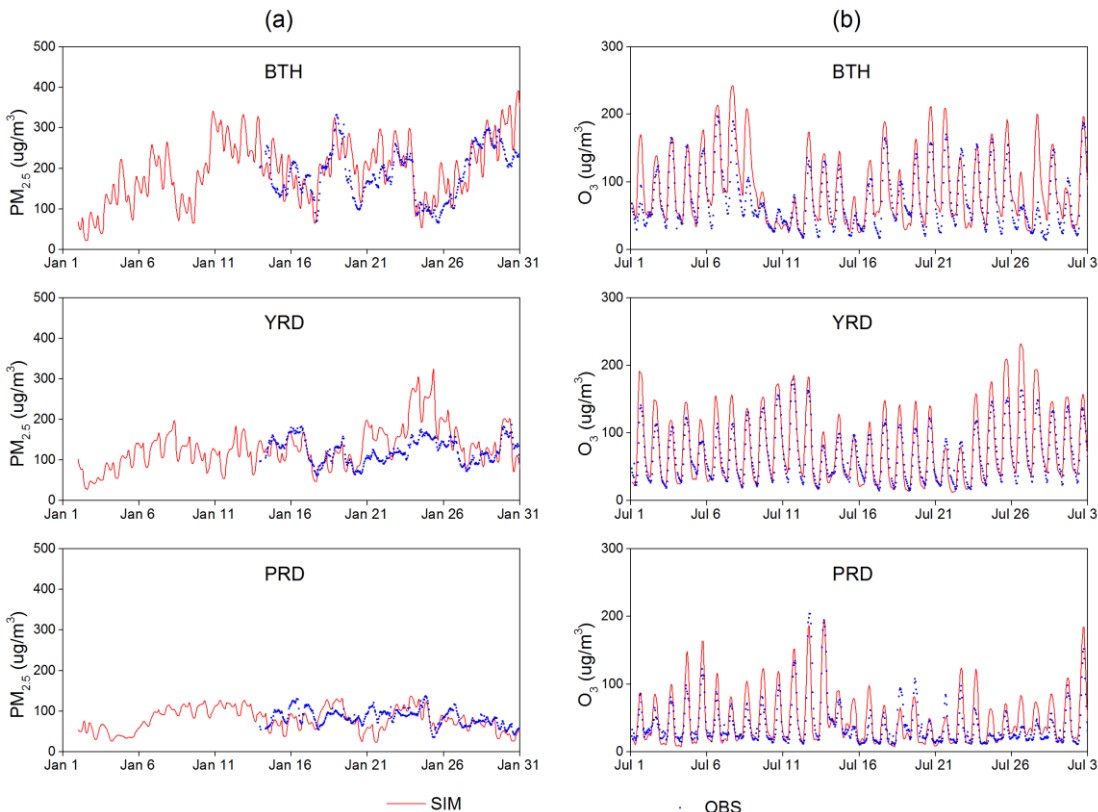

**Figure 8. Time series of hourly concentrations of (a) PM$_{2.5}$ in January and (b) O$_3$ in July 2013, under the short-term air quality application over three key regions in China: the Beijing-Tianjin-Hebei area (BTH), the Yangtze River Delta (YRD), and the Pearl River Delta (PRD). The hourly concentrations in each region were derived by averaging all monitoring stations located in the region.**

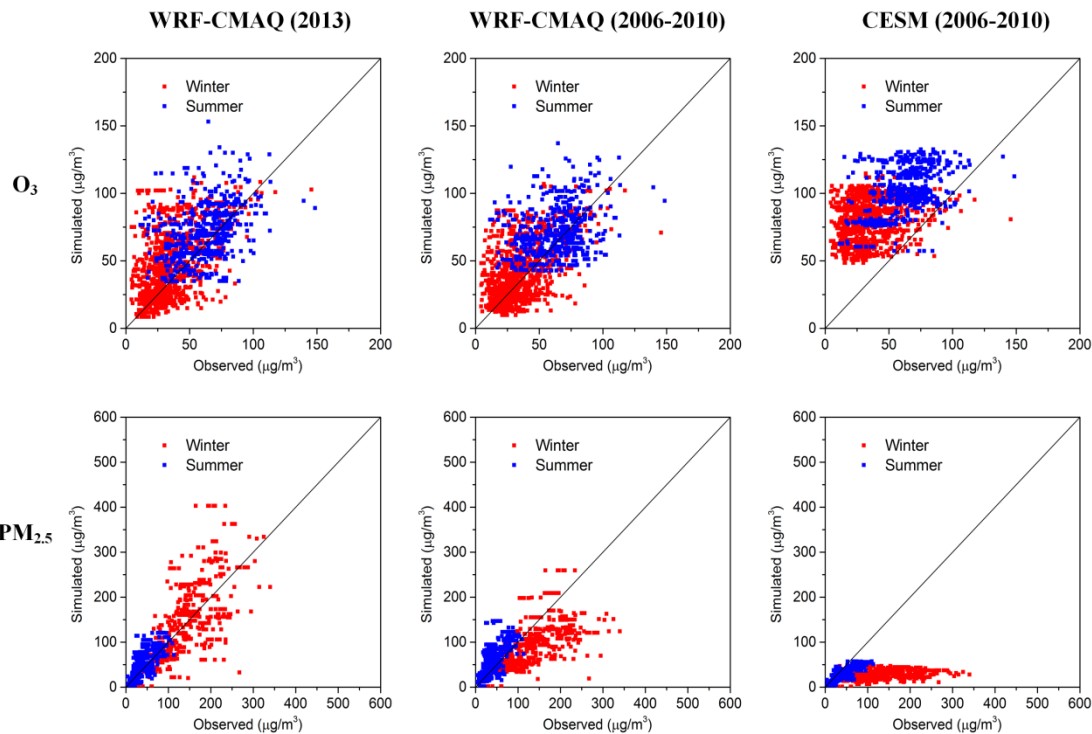

**Figure 9. Scatter plots of simulated (by WRF-CMAQ (left and middle) and CESM (right)) and observed PM$_{2.5}$ (bottom) and O$_3$ (top) in winter (red) and summer (blue). Each scatter represents the value at each observational site. The years of observational data, WRF-CMAQ (2013), WRF-CMAQ (2006-2010), and CESM simulations were 2013, 2013, 2006-2010, and 2006-2010, respectively. Note that for O$_3$ observed data in winter, observational data in year 2014 were used because of some errors in the data in year 2013.**

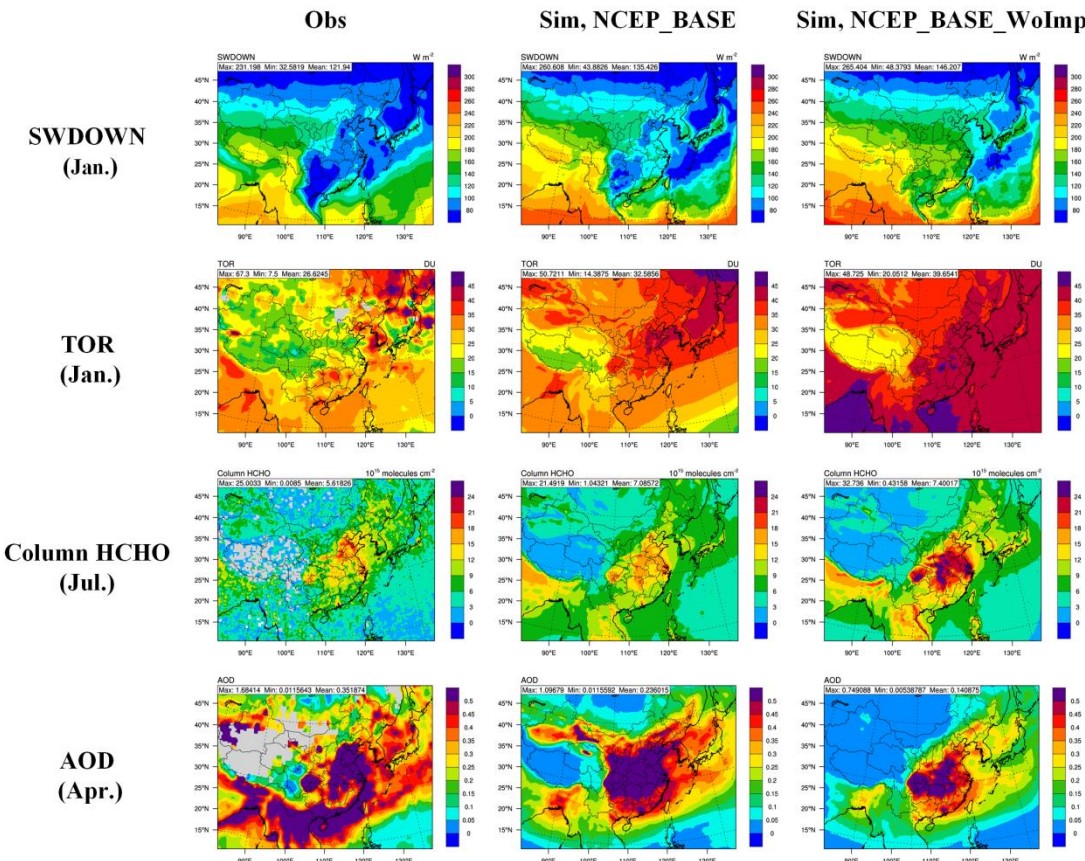

**Figure 10. Comparison of spatial distributions of SWDOWN in January, TOR in January, column HCHO in July, and AOD in April 2013 from: (a) satellite observations, (b) baseline simulation (NCEP_BASE: with aerosol feedbacks, using CESM-derived BCs, BEIS3 emissions, and revised dust scheme), and (c) sensitivity simulation (NCEP_BASE_WoImp: without aerosol feedbacks, using fixed BCs, MEGAN emissions, and default dust scheme).**

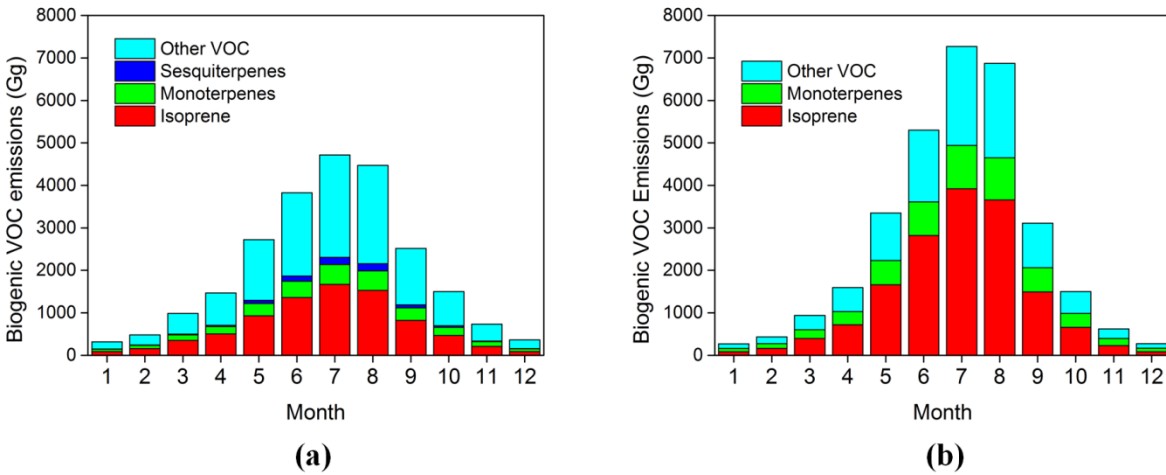

**Figure 11. Biogenic VOC emissions over China in 2013 estimated by (a) BEIS and (b) MEGAN.**

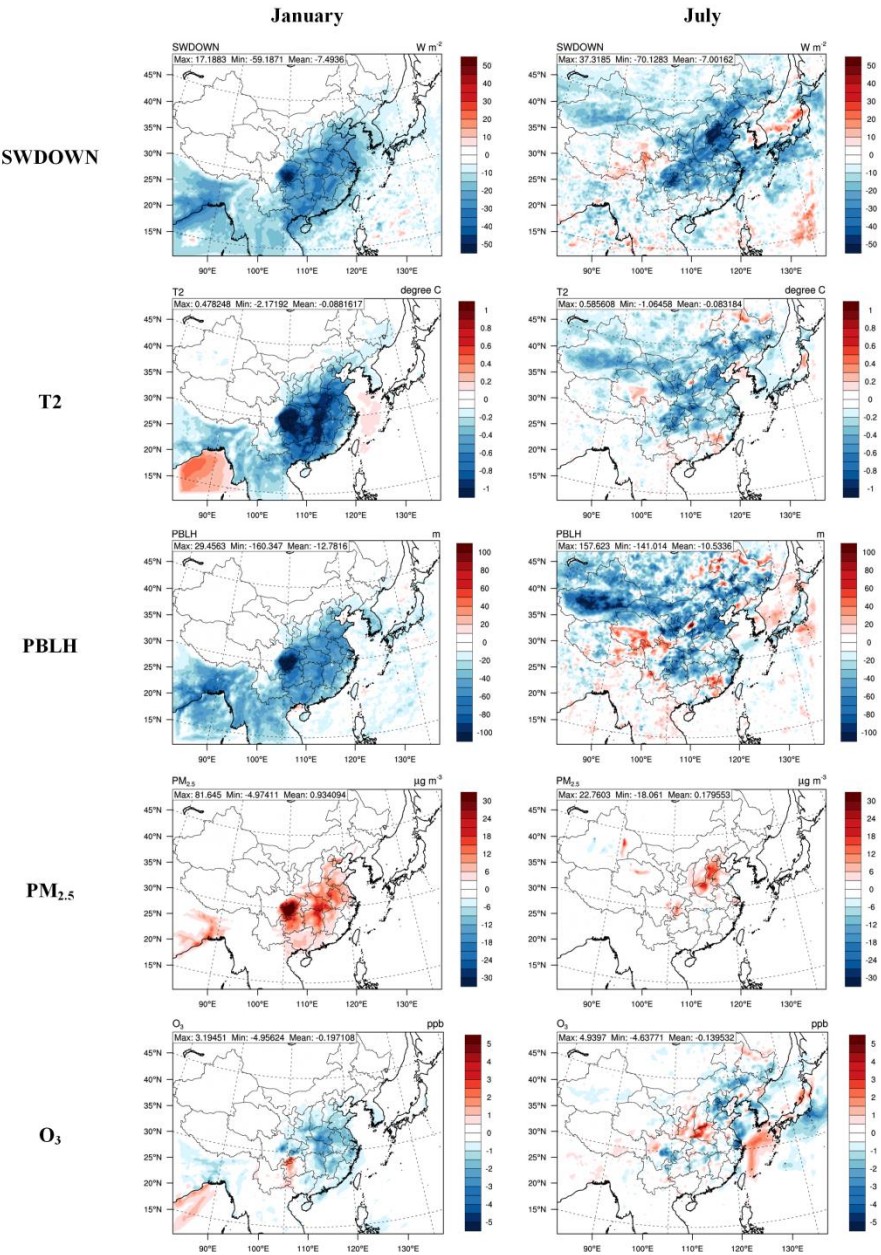

**Figure 12. Effects of aerosol feedbacks on net shortwave flux at the surface (SWDOWN), T2, planetary boundary layer height (PBLH), surface PM$_{2.5}$ and O$_3$ concentrations in January 2008 and in July 2007. The results are from the differences between the feedback and no feedback configurations (i.e., CESM_BASE: with feedback and CESM_BASE_Sens: without feedback) of the two-way coupled WRF-CMAQ simulations.**