# Peer review of "Multi-year Downscaling Application of Two-Way Coupled WRF3.4 and CMAQv5.0.2 over East Asia for Regional Climate and Air Quality Modeling: Model Evaluation and Aerosol Direct Effects"

_Geoscientific Model Development, 2016_

## Short Comment (SC1) · 12 Dec 2016

Dear authors,

in my role as Executive editor of GMD, I would like to bring to your attention our Editorial version 1.1:

http://www.geosci-model-dev.net/8/3487/2015/gmd-8-3487-2015.html

This highlights some requirements of papers published in GMD, which is also available on the GMD website in the 'Manuscript Types' section:

http://www.geoscientific-model-development.net/submission/manuscript_types.html

In particular, please note that for your paper, the following requirement has not been met in the Discussions paper:

- "The main paper must give the model name and version number (or other unique identifier) in the title."

Please add the version numbers of WRF and CMAQ in the title upon your revised submission to GMD. Anyhow, the dash in WRF-CMAQ seem to be missing in the title, which is inconsitent with the rest of the article.

Yours,

Astrid Kerkweg

———————————————

---

## Referee Comment (RC1) · Anonymous Referee #1 · 17 Jan 2017

This study concerns the application and evaluation of a regional climate-chemistry modeling system. This is certainly an interesting topic and represents an advancement in climate-chemistry modeling. The evaluation of the system for both climate model driven simulations and re-analysis driven simulations are reasonably thorough and successful. My primary criticism of this paper is the lack of detailed description of the GCM model, the downscaling, regional model configuration, and execution. Even though references are given for the CESM modeling and the chemical and aerosol processes are briefly described I would like to see further description of the CESM physics, spin-up, constraints, etc. I do not understand how this represents a climate scenario

when it is for past years and is evaluated against observations. What does RCP4.5 for these years represent. Do these runs use observation based SSTs? If these runs were spun-up from pre-industrial times without any observed data constraints, there would be no reason to expect agreement with observations. If bias corrections are made to both the meteorology and chemistry, then how do these runs substantially differ from re-analysis driven runs? Please explain the rationale and expectations of these runs.

I'm also wondering about data assimilation in WRF. Our experience has been that long runs of WRF (one month or longer) need some sort of DA or frequent re-initialization. If not in this case, how were the meteorology statistics this good? Even downscaling from GCMs often use data assimilation from the GCM. Also, an important omission from the WRF physics description is the LSM. Overall, I think that this study is worthy of reporting in GMD, especially the sensitivities of AQ and meteorology to aerosol direct radiative effects, and also the effects of dynamic BCs and biogenic and dust emissions. However, more explanation and description is needed particularly to help the reader understand the significance of the climate runs.

Specific comments:

P4Ins21-22: This statement about "correcting the roughness length by increasing the friction velocity by 1.5 times when calculating wind speeds in the ACM2 PBL scheme to reduce the overpredictions of wind speeds" needs more explanation. First, if the roughness lengths need correcting why not change them and not the friction velocity. Second, what is the problem with roughness lengths? How are they specified and what are they? Our experience has not shown general overpredictions in windspeed. Windspeed and friction velocity are strongly affected by the LSM and surface layer scheme which are not even mentioned here. Also the LU scheme and data are important. The USGS 24cat data is way out of date especially for China where urbanization has been dramatic. Why not use MODIS LU?

P5In11-12: Why not use same vertical structure for WRF-CMAQ as CESM?

P5In27: what is TOR?

Page 6: I don't understand what is the point of using RCP projections when modeling retrospectively. It seems that 2008 emission inventories are used for more detailed spatial-temporal allocation. Then why not just use these inventories? What is an RCP projection for past years? Please explain the logic here.

P8In1-2: Should also report RMSE or MAE. Small biases don't tell whole story. Large over and under predictions could cancel out.

P8In15-16: saying that large errors could be attributable to KF and Morrison schemes is pretty meaningless.

P8In28: Are the results shown in Fig5 averages for all 5 years?

P9In16: what are "upper BCs"? and where do they come from? And why are they particularly uncertain?

P9In16-17: Another meaningless statement about uncertainties in about everything possibly causing errors in NO2 column. Can you provide more insightful analyses?

P9In28-30: Please clarify this sentence.

P10In23-24: If a figure is important enough to be discussed in the text (S2) it should be in the main paper and not in the supplement. The reader should not need to see the supplement to follow the discussion.

P11In12-13: The names of the simulations are confusing. The "baseline" is NCEP\_BASE\_Imp but the sensitivity is NCEP\_BASE which sounds more like it should be the base.

P11In25-26: How are the fixed BCs derived?

P12In4: S4 should be in main paper.

P12In15: "close" should be "closer"

P13In11-12: Aerosol effects on photolysis in CMAQ do not depend on aerosol feedback in the WRF-CMAQ system. The more likely cause for ozone decline in the feedback run is increased NOx titration in cities due to reduced PBL mixing. Table 1: what LSM and surface layer scheme? Table 3 and 4: Better to have un-normalized error for T2, RH2, WS10, WD10

---

## Referee Comment (RC2) · Anonymous Referee #2 · 27 Jan 2017

This study reports the evaluation against measurements of the output from a dynamical downscaling link between the global Community Earth System Model (CESM) and the WRF-CMAQ modelling system over the East Asia region for a number of meteorological and air quality composition variables. The climatological simulations were for RCP4.5 for 2006-10 and the air quality applications were for winter and summer months in 2013 (principal compositional variables of interest: PM2.5 and O3). The authors report satisfactory prediction of major meteorological variables, although see the first of the general comments below. The paper reports on a major piece of work, with what appear to be generally appropriate methods, and is within the scope for

consideration of publication in GMD.

General comments

(1) The description of the downscaling (P5-6) indicates that aspects of it involves significant bias corrections, so to what extent is it valid to judge model performance by model-observation statistics? For example, it is stated on P8, lines 1-6, that the improved statistical performance of the modelling approach used in this study may be related to the bias-correction applied. If a bias correction is applied then presumably we expect better model-observation statistics, so have we learned anything fundamental about the model performance by these comparison statistics?

(2) The model-observation statistics should include RMSE instead of, or in place of, the normalized mean error (NME). The former is the statistic usually used alongside the correlation coefficient and mean bias (or normalised mean bias) in the suite of statistics that captures the spectrum of model performance characteristics.

(3) In general, the discussion of model output against meteorological and compositional variations is (i) vague, i.e. non-quantitative (using phrasing like agreed well, satisfactory, etc.), and (ii) lacking explanatory insight, i.e. lists of potential reasons for discrepancy are given which could be written down as potential explanations without needing to do these comparisons. The authors should endeavour to provide more quantitative assessments of model performance, including how their mod-obs statistics compare with expectation and with other studies, and also to provide some more informed analysis of what is the driving explanation for mod-obs discrepancies for particular variables or circumstances.

Specific comments

P1, L27: The phrasing "The model showed good ability to predict PM2.5 . . ..and O3. . ." is non-quantitative and vague.

P4, L20: Rephrase as "Several modifications in model. . ."

P7, L16: Although the acronym TOR is defined here, there needs to be some further explanation of what it means in practice, particularly in the context of its relevance to model performance evaluation.

P12, L18: "were much closer to...
* * *

---

## Author Comment (AC1) · 27 Mar 2017

**Executive editor**

*In my role as Executive editor of GMD, I would like to bring to your attention our Editorial version 1.1: http://www.geosci-model-dev.net/8/3487/2015/gmd-8-3487-2015.html.*

*This highlights some requirements of papers published in GMD, which is also available on the GMD website in the 'Manuscript Types' section:*

*http://www.geoscientific-model-development.net/submission/manuscript_types.html*

*In particular, please note that for your paper, the following requirement has not been met in the Discussions paper: "The main paper must give the model name and version number (or other unique identifier) in the title."*

*Please add the version numbers of WRF and CMAQ in the title upon your revised submission to GMD. Anyhow, the dash in WRF-CMAQ seem to be missing in the title, which is inconsitent with the rest of the article.*

**Response:** The version numbers of WRF and CMAQ have been added to the paper title. We also changed "online coupled" to "two-way coupled" in the title for accuracy (Wong et al., 2012).

**Reference**

Wong, D. C., Pleim, J., Mathur, R., Binkowski, F., Otte, T., Gilliam, R., Pouliot, G., Xiu, A., Young, J. O., and Kang, D.: WRF-CMAQ two-way coupled system with aerosol feedback: software development and preliminary results, Geosci. Model Dev., 5, 299–312, doi:10.5194/gmd-5-299-2012, 2012.

---

## Author Comment (AC3) · 27 Mar 2017

*This study reports the evaluation against measurements of the output from a dynamical downscaling link between the global Community Earth System Model (CESM) and the WRF-CMAQ modelling system over the East Asia region for a number of meteorological and air quality composition variables. The climatological simulations were for RCP4.5 for 2006-10 and the air quality applications were for winter and summer months in 2013 (principal compositional variables of interest: PM2.5 and O3). The authors report satisfactory prediction of major meteorological variables, although see the first of the general comments below. The paper reports on a major piece of work, with what appear to be generally appropriate methods, and is within the scope for consideration of publication in GMD.*

**Response:** We thank Referee #2 for the constructive comments. Please see below our point-by-point replies to other comments.

General comments
*(1) The description of the downscaling (P5-6) indicates that aspects of it involves significant bias corrections, so to what extent is it valid to judge model performance by model-observation statistics? For example, it is stated on P8, lines 1-6, that the improved statistical performance of the modelling approach used in this study may be related to the bias-correction applied. If a bias correction is applied then presumably we expect better model-observation statistics, so have we learned anything fundamental about the model performance by these comparison statistics?*

**Response:** While using bias-corrected ICs/BCs does improve WRF-CMAQ's model performance, it does not make model-observation comparison invalid. While meteorological reanalysis data were used to correct biases in meteorological ICs/BCs based on CESM-NCSU's results and satellite retrievals of $O_3$ were used to constrain their upper boundary conditions, observational data were used for model performance evaluation. Because GCMs generally suffer from systematic biases to a certain extent, bias correction to the GCM (i.e., CESM) boundary conditions was applied in this study to improve the model performance in simulating regional climate. By comparing to the traditional approach without GCM bias corrections, previous studies (Xu and Yang, 2012; Bruyère et al., 2014; Done et al., 2015) have shown that the improved dynamical downscaling method with GCM bias corrections greatly improves the downscaled climate. The bias-correction technique is also used in the NCAR CESM global bias-corrected CMIP5 output to support WRF/MPAS research (https://rda.ucar.edu/datasets/ds316.1/). Also note that the bias correction is applied to the ICs/BCs, rather than the model results. So, the model-observation comparison will provide insights into the model's capability in capturing observations.

*(2) The model-observation statistics should include RMSE instead of, or in place of, the normalized mean error (NME). The former is the statistic usually used alongside the correlation coefficient and mean bias (or normalised mean bias) in the suite of statistics that captures the spectrum of model performance characteristics.*

**Response:** As suggested, we have added the root mean square error (RMSE) in the statistics tables (Table 3, 4 and S3) in place of the normalized mean error (NME), and added the mean absolute gross

error (MAGE) in Table 3. The model performed well for T2 and RH2, with MBs of -0.6 ℃ and 0.8%, correlation coefficients of 0.97 and 0.72, MAGEs of 2.4 ℃ and 9.7%, and RMSEs of 3.2 ℃ and 12.6%, respectively. WS10 was moderately overpredicted by 22.2%, with an MB of 0.6 m/s, an MAGE of 1.2 m/s and a RMSE of 1.6 m/s. We have added this in the Section 3.1 of the revised manuscript.

*(3) In general, the discussion of model output against meteorological and compositional variations is (i) vague, i.e. non-quantitative (using phrasing like agreed well, satisfactory, etc.), and (ii) lacking explanatory insight, i.e. lists of potential reasons for discrepancy are given which could be written down as potential explanations without needing to do these comparisons. The authors should endeavour to provide more quantitative assessments of model performance, including how their mod-obs statistics compare with expectation and with other studies, and also to provide some more informed analysis of what is the driving explanation for mod-obs discrepancies for particular variables or circumstances.*

**Response:** As suggested, we have provided more quantitative assessments of model performance in terms of MB, NMB, or RMSE in the abstract, result and conclusion sections of the revised manuscript. The model biases or errors can be attributed to many factors. Pinpointing the exact causes is not a trivial effort, often involving large amounts of sensitivity simulations and in some cases, model further development and improvement that are not permitted with our very limited resources. Nevertheless, we have provided more insights into the model's performance statistics and how they are compared with other studies, wherever possible. For example, we have compared the performance of several meteorological variables with the benchmarks suggested by Emery et al. (2001) in the Section 3.1 of the manuscript. Emery et al. (2001) suggested the benchmarks for satisfactory performance for T2 (MB within ±0.5 ℃, MAGE of ≤ 2.0 ℃) and WS10 (MB within ±0.5 m/s, MAGE and RMSE of 2.0 m/s). In the climatological application, the MB and MAGE of T2 and the MB of WS10 are close to the benchmark, the MAGE and RMSE of WS10 are within the benchmark, and hence the performance is deemed acceptable.

We have also compared the CMAQ performance of chemical predictions in this study with other studies, as shown in the Section 3.2 of the manuscript. The revised text is as follows:

The CMAQ performance of chemical predictions in this study was comparable to or even better than those of other air quality studies over East Asia (Wang et al., 2009; 2012; Liu et al., 2010; Zheng et al., 2015; Hu et al., 2016; Liu et al., 2016; Zhang et al., 2016a). This study predicted relatively well for most chemical species in most months. Compared with other regional modeling studies, WRF-CMAQv5.0.2 used in this study outperformed MM5/CMAQv4.6, which tend to underpredict the surface concentrations of major species with NMBs generally greater than -40% and overpredict surface $O_3$ concentrations in most months with NMBs generally higher than 20% over East Asia according to the evaluation results of Zhang et al. (2016a). A relatively good performance of CMAQv5.0.1 was also reported by Hu et al. (2016). Global models such as GEOS-Chem and CESM tend to underpredict $PM_{2.5}$ concentrations (by about -50% as reported by Jiang et al., 2013) and overpredict $O_3$ concentrations (by about 50% as reported by He and Zhang, 2014; Wang et al., 2013) in China/East Asia because of relatively coarse grid resolution and limitations in some model treatments (e.g., missing emissions of unspeciated primary $PM_{2.5}$, and discrepancies in surface layer height and vertical mixing).

Specific comments

*P1, L27: The phrasing "The model showed good ability to predict PM2.5 . . ..and O3. . ." is non-quantitative and vague.*

**Response:** The above sentence has been revised to include more quantitative assessment as follows:

The model showed good ability to predict $PM_{2.5}$ in winter (with a normalized mean bias (NMB) of 6.4% in 2013) and $O_3$ in summer (with an NMB of 18.2% in 2013) in terms of statistical performance and spatial distributions.

In addition, we have added this in the abstract of the revised manuscript.

*P4, L20: Rephrase as "Several modifications in model. . ."*

**Response:** Revised as suggested.

*P7, L16: Although the acronym TOR is defined here, there needs to be some further explanation of what it means in practice, particularly in the context of its relevance to model performance evaluation.*

**Response:** TOR represents tropospheric ozone residual or column abundance of $O_3$. We have clarified this in the Section 2.2 of the revised manuscript.

*P12, L18: "were much closer to. . .*

**Response:** Revised as suggested.

**Reference**

Bruyere, C. L., Done, J. M., Holland, G. J., and Fredrick, S.: Bias corrections of global models for regional climate simulations of high-impact weather, Clim. Dynam., 43, 1847–1856, doi:10.1007/s00382-013-2011-6, 2014.

Done, J. M., Holland, G. J., Bruyere, C. L., Leung, L. R., and Suzuki-Parker, A.: Modeling high-impact weather and climate: lessons from a tropical cyclone perspective, Climatic Change, 129, 381–395, doi:10.1007/s10584-013-0954-6, 2015.

Emery, C., Tai, E., and Yarwood, G.: Enhanced meteorological modeling and performance evaluation for two texas episodes, Report to the Texas Natural Resources Conservation Commission, prepared by ENVIRON, International Corp., Novato, CA, available at: http://www.tceq.state.tx.us/assets/public/implementation/air/am/contracts/reports/mm/EnhancedMetMo delingAndPerformanceEvaluation.pdf, 2001.

He, J., and Zhang, Y.: Improvement and further development in CESM/CAM5: gas-phase chemistry and inorganic aerosol treatments, Atmos. Chem. Phys., 14, 9171–9200, doi:10.5194/acp-14-9171-2014, 2014.

Hu, J., Chen, J., Ying, Q., and Zhang, H.: One-year simulation of ozone and particulate matter in China using WRF/CMAQ modeling system, Atmos. Chem. Phys., 16, 10333–10350, doi:10.5194/acp-16-10333-2016, 2016.

Jiang, H., Liao, H., Pye, H. O. T., Wu, S., Mickley, L. J., Seinfeld, J. H., and Zhang, X. Y.: Projected effect of 2000-2050 changes in climate and emissions on aerosol levels in China and associated transboundary transport, Atmos. Chem. Phys., 13, 7937–7960, doi:10.5194/acp-13-7937-2013, 2013.

Liu, X., Zhang, Y., Cheng, S., Xing, J., Zhang, Q., Streets, D. G., Jang, C., Wang, W., and Hao, J.: Understanding of regional air pollution over China using CMAQ, part I performance evaluation and seasonal variation, Atmos. Environ., 44, 2415–2426, doi:10.1016/j.atmosenv.2010.03.035, 2010.

Liu, X., Zhang, Y., Zhang, Q., and He, M.: Application of online-coupled WRF/Chem-MADRID in East Asia: Model evaluation and climatic effects of anthropogenic aerosols, Atmos. Environ., 124, 321–336, doi:10.1016/j.atmosenv.2015.03.052, 2016.

Wang, K., Zhang, Y., Jang, C., Phillips, S., and Wang, B.: Modeling intercontinental air pollution transport over the trans-Pacific region in 2001 using the Community Multiscale Air Quality modeling system, J. Geophys. Res. Atmos., 114, D04307, doi:10.1029/2008JD010807, 2009.

Wang, K., Zhang, Y., Nenes, A., and Fountoukis, C.: Implementation of dust emission and chemistry into the Community Multiscale Air Quality modeling system and initial application to an Asian dust storm episode, Atmos. Chem. Phys., 12, 10209–10237, doi:10.5194/acp-12-10209-2012, 2012.

Wang, Y., Shen, L., Wu, S., Mickley, L., He, J., and Hao, J.: Sensitivity of surface ozone over China to 2000-2050 global changes of climate and emissions, Atmos. Environ., 75, 374–382, doi:10.1016/j.atmosenv.2013.04.045, 2013.

Xu, Z., and Yang, Z.: An Improved Dynamical Downscaling Method with GCM Bias Corrections and Its Validation with 30 Years of Climate Simulations, J. Climate, 25, 6271–6286, doi:10.1175/JCLI-D-12-00005.1, 2012.

Zhang, Y., Zhang, X., Wang, L., Zhang, Q., Duan, F., and He, K.: Application of WRF/Chem over East Asia: Part I. Model evaluation and intercomparison with MM5/CMAQ, Atmos. Environ., 124, 285–300, doi:10.1016/j.atmosenv.2015.07.022, 2016a.

Zheng, B., Zhang, Q., Zhang, Y., He, K. B., Wang, K., Zheng, G. J., Duan, F. K., Ma, Y. L., and Kimoto, T.: Heterogeneous chemistry: a mechanism missing in current models to explain secondary inorganic

aerosol formation during the January 2013 haze episode in North China, Atmos. Chem. Phys., 15, 2031–2049, doi:10.5194/acp-15-2031-2015, 2015.

---

## Author Response (AR1)

**Executive editor**

*In my role as Executive editor of GMD, I would like to bring to your attention our Editorial version 1.1:*
*http://www.geosci-model-dev.net/8/3487/2015/gmd-8-3487-2015.html.*
*This highlights some requirements of papers published in GMD, which is also available on the GMD*
*website in the 'Manuscript Types' section:*
*http://www.geoscientific-model-development.net/submission/manuscript_types.html*
*In particular, please note that for your paper, the following requirement has not been met in the*
*Discussions paper: "The main paper must give the model name and version number (or other unique*
*identifier) in the title."*
*Please add the version numbers of WRF and CMAQ in the title upon your revised submission to GMD.*
*Anyhow, the dash in WRF-CMAQ seem to be missing in the title, which is inconsitent with the rest of*
*the article.*

**Response:** The version numbers of WRF and CMAQ have been added to the paper title. We also changed "online coupled" to "two-way coupled" in the title for accuracy (Wong et al., 2012).

**Anonymous Referee #1**

*This study concerns the application and evaluation of a regional climate-chemistry modeling system. This is certainly an interesting topic and represents an advancement in climate-chemistry modeling. The evaluation of the system for both climate model driven simulations and re-analysis driven simulations are reasonably thorough and successful.*

**Response:** We thank Referee #1 for the positive comments. Please see below our point-by-point replies to other comments.

*My primary criticism of this paper is the lack of detailed description of the GCM model, the downscaling, regional model configuration, and execution. Even though references are given for the CESM modeling and the chemical and aerosol processes are briefly described I would like to see further description of the CESM physics, spin-up, constraints, etc. I do not understand how this represents a climate scenario when it is for past years and is evaluated against observations. What does RCP4.5 for these years represent. Do these runs use observation based SSTs? If these runs were spun-up from pre-industrial times without any observed data constraints, there would be no reason to expect agreement with observations. If bias corrections are made to both the meteorology and chemistry, then how do these runs substantially differ from re-analysis driven runs? Please explain the rationale and expectations of these runs.*

**Response:** The CESM-NCSU model development, application, and evaluation have been published in several journal papers (e.g., He and Zhang, 2014; He et al., 2015a, b; Gantt et al., 2014; Glotfelty et al., 2017a, b; Glotfelty and Zhang, 2017). Since CESM-NCSU has been well documented, it is a common practice for us to cite those references rather than repeat the CESM-NCSU model description in our paper. To address the reviewer's comment, we have added a brief description on the CESM-NCSU's configuration, initial conditions and the application mode in Section 2.2. We have also included a Table (Table S1) in the supplementary material to summarize the model configuration including physical schemes and chemical options used in CESM-NCSU applications under the RCP scenarios. More detailed descriptions can be found in He and Zhang (2014) and Glotfelty et al. (2017a, b).

The CESM-NCSU model has been applied for decadal global climate and air quality predictions to simulate the "current" climate (2001–2010) and the "future" climate (2046-2055) driven with the RCPs emissions for both the current and future decades (Glotfelty et al., 2017a and Glotfelty and Zhang, 2017). The CESM simulation for 2001–2010 is performed with fully-coupled CESM with CESM1.2.2 B_2000_STRATMAM7_CN configuration (rather than using prescribed SST), which represents a fully-coupled CESM configuration including prognostic simulation of the atmosphere, ocean, land, and sea ice from the various component models.

Global climate/chemistry models applied at a coarse spatial resolution may not well resolve mesoscale features over a regional domain of interest or well predict local air quality and thus are not suitable for high-resolution regional climate, air quality and health impact studies. Therefore, we have planned to downscale CESM runs with a regional model, which is the two-way coupled WRF-CMAQ for both a current period (2006-2010) and a future period (2046-2050) to study the impacts of projected changes in climate and anthropogenic emissions under the RCP4.5 scenario. In this paper, multi-year downscaling applications from CESM-NCSU simulations under RCP 4.5 were conducted to simulate

regional climate and air quality in current year period (2006-2010) and evaluated against observations during this 5-yr period, which is the first part of the study. The results for future years will be presented in a future paper. The results from this Part I paper will establish a baseline for a future Part II paper. The WRF-CMAQ simulations are driven with CESM-NCSU downscaling data under RCP 4.5, and the projected emissions for 2046-2050 WRF-CMAQ simulations are based on MIX2008 and RCP4.5, so the downscaling simulations during 2006-2010 represent multi-year climatological baseline simulations under RCP4.5, and they will be further used to investigate future regional climate and air quality change in a future paper. To avoid confusion, we have revised the paper to clarify the above points in several places including abstract, Section 1, and conclusion.

This study presents the first application and evaluation of the two-way coupled WRF-CMAQ model for multi-year climatological simulations using the dynamical downscaling technique. Because GCMs generally suffer from systematic biases to a certain extent, bias correction to the GCM (i.e., CESM) initial and boundary conditions was applied in this study to improve the model performance in simulating regional climate. The bias-correction method corrected the mean climatological biases in temperature, water vapor, geopotential height, wind, and soil moisture variables using the NCEP reanalysis data following the approach of Xu and Yang (2012), and allowed the retention of the CESM-NUSU simulated climatic changes in the mean seasonal state, diurnal cycle, and variance of inter-annual variation. The bias-correction method used for the initial and boundary conditions derived from CESM-NCSU is described in Yahay et al. (2016). As described in Section 2.2, in this bias-correction approach, monthly climatological averages for ICs and BCs are first derived from both NCEP and CESM_NCSU cases. The differences between the ICs and BCs from the NCEP and CESM_NCSU climatological averages are then added onto the CESM_NCSU ICs and BCs to generate bias-corrected CESM_NCSU ICs/BCs. WRF-CMAQ simulations using bias-corrected meteorological ICs/BCs from CESM-NCSU are therefore different from the simulations using the NCEP reanalysis for meteorological ICs/BCs. The bias-correction method corrected the major biases in the meteorological variables that can cause serious issues for regional climate downscaling while retaining climate variability within the GCM for both current and future simulations. So we do not expect the climatological runs achieve the same performance as the re-analysis driven runs. Note that previous studies (Xu and Yang, 2012; Bruyère et al., 2014; Done et al., 2015) have shown that the improved dynamical downscaling method with GCM bias corrections greatly improves the downscaled climate. The bias-correction technique is also used in the NCAR CESM global bias-corrected CMIP5 output to support WRF/MPAS research (https://rda.ucar.edu/datasets/ds316.1/).

*I'm also wondering about data assimilation in WRF. Our experience has been that long runs of WRF (one month or longer) need some sort of DA or frequent re-initialization. If not in this case, how were the meteorology statistics this good? Even downscaling from GCMs often use data assimilation from the GCM. Also, an important omission from the WRF physics description is the LSM.*

**Response:** In order to simulate regional meteorology as accurately as possible and preserve the chemistry–meteorology feedbacks, re-initialization in WRF was used in the multi-year climatological application. The climatological simulations were reinitialized every 15 days in this work, which provides a compromise to allow the simulation of mesoscale features and aerosol feedbacks while periodically constraining the meteorological fields not significantly deviated from the GCM. Qian et al. (2003) found that frequent re-initialization with frequencies of 10 days to 1 month improved the

accuracy in regional climate downscaling. Data assimilation in WRF was not used to allow chemistry–meteorology feedbacks within the system. We have clarified this in the Section 2.1 of the revised manuscript. The land surface model is the Pleim–Xiu land surface model. We have added them into Table 1.

*Overall, I think that this study is worthy of reporting in GMD, especially the sensitivities of AQ and meteorology to aerosol direct radiative effects, and also the effects of dynamic BCs and biogenic and dust emissions. However, more explanation and description is needed particularly to help the reader understand the significance of the climate runs.*

**Response:** We have added more explanation and description to help the readers understand the significance of the climate runs. Please refer to the above responses.

Specific comments:

*P4lns21-22: This statement about "correcting the roughness length by increasing the friction velocity by 1.5 times when calculating wind speeds in the ACM2 PBL scheme to reduce the overpredictions of wind speeds" needs more explanation. First, if the roughness lengths need correcting why not change them and not the friction velocity. Second, what is the problem with roughness lengths? How are they specified and what are they? Our experience has not shown general overpredictions in windspeed. Windspeed and friction velocity are strongly affected by the LSM and surface layer scheme which are not even mentioned here. Also the LU scheme and data are important. The USGS 24cat data is way out of date especially for China where urbanization has been dramatic. Why not use MODIS LU?*

**Response:** Large overpredictions in WS10 with NMBs of 48.7%-101.0% from WRF simulations have been reported in the literature (Penrod et al., 2014; Cai et al., 2016; Zhang et al., 2016a) because of unresolved subgrid-scale topographic features and uncertainties in parameterizations of turbulent fluxes in WRF (Hanna and Yang, 2001; Rontu, 2006; Mass and Ovens, 2011). The overpredictions in WS10 are likely caused by low surface drag due to the inappropriate representation of surface roughness because the detailed surface structure cannot be reproduced at a coarse grid resolution of 36-km. However, a rigorous surface roughness correction algorithm is not available in WRF v3.4 that is used in the two-way coupled WRF-CMAQ. To correct the WS10 bias, following Mass and Ovens (2010) and our previous studies (Zheng et al., 2015; Zhang et al., 2016b), a highly simplified indirect correction method is used in this study, namely, the surface drag is increased by 1.5 times (which is applied to the friction velocity) when calculating wind speeds in the ACM2 PBL scheme. The simple wind correction method effectively reduces the overpredictions of wind speeds. To address the reviewer's concern, we have indicated the highly simplified wind bias correction method as a limitation of this work in the conclusion section. A more rigorous method should be used for future work.

The USGS 24-category land use data is indeed way out of date for China where urbanization has been dramatic, which would also partly contribute to the overprediction in WS10. We have indicated this in the Section 3.1 of the revised manuscript.

We used Pleim–Xiu land surface model (PX-LSM, Xiu and Pleim, 2001) and Pleim–Xiu surface layer scheme. For best consistency between the WRF and CMAQ model, the PX-LSM and the ACM2 PBL scheme were used in the two-way coupled WRF-CMAQ model (https://www.airqualitymodeling.org/index.php/CMAQ_version_5.1_(November_2015_release)_Techn

ical_Documentation). We have added them into Table 1.

*P5ln11-12: Why not use same vertical structure for WRF-CMAQ as CESM?*

**Response:** The vertical coordinate in CESM is a hybrid sigma-pressure system, which is different from the WRF sigma coordinate. Thus we do not use the same vertical structure for WRF-CMAQ as CESM. We have clarified this in the Section 2.2 of the revised manuscript.

*P5ln27: what is TOR?*

**Response:** TOR represents tropospheric ozone residual. We have clarified this in the Section 2.2 of the revised manuscript.

*Page 6: I don't understand what is the point of using RCP projections when modeling retrospectively. It seems that 2008 emission inventories are used for more detailed spatial-temporal allocation. Then why not just use these inventories? What is an RCP projection for past years? Please explain the logic here.*

**Response:** RCP emissions are available for current and future decadal periods. The CESM-NCSU model has been recently applied for decadal global climate and air quality predictions to simulate the "current" climate scenario (2001–2010) and the "future" climate scenario (2046-2055) driven with the RCPs emissions for both current and future decades (Glotfelty et al., 2017a), therefore those "current" and "future" simulations represent multi-year climatological simulations under RCPs. The regional climatological simulations were driven with CESM-NCSU downscaling data under RCP 4.5. In order to achieve better performance for the regional WRF-CMAQ simulation, the MIX 2008 emission inventory is used for current years, and the emissions of some sectors that were not available from MIX 2008 were taken from RCP4.5. As we explained above, this paper only focuses on current year simulations which will be used as a baseline simulation for a future paper. We have clarified this in the Section 2.3 of the revised manuscript.

*P8ln1-2: Should also report RMSE or MAE. Small biases don't tell whole story. Large over and under predictions could cancel out.*

**Response:** As suggested, we have added the root mean square error (RMSE) in the statistics tables (Table 3, 4 and S3) in place of the normalized mean error (NME), and added the mean absolute gross error (MAGE) in Table 3. The model performed well for T2 and RH2, with MBs of -0.6 ℃ and 0.8%, correlation coefficients of 0.97 and 0.72, MAGEs of 2.4 ℃ and 9.7%, and RMSEs of 3.2 ℃ and 12.6%, respectively. WS10 was moderately overpredicted by 22.2%, with an MB of 0.6 m/s, an MAGE of 1.2 m/s and a RMSE of 1.6 m/s. Emery et al. (2001) suggested the benchmarks for satisfactory performance for T2 (MB within ±0.5 ℃, MAGE of ≤ 2.0 ℃) and WS10 (MB within ±0.5 m/s, MAGE and RMSE of 2.0 m/s). In the climatological application, the MB and MAGE of T2 and the MB of WS10 are close to the benchmark, the MAGE and RMSE of WS10 are within the benchmark, and hence the performance is deemed acceptable. We have added this in the Section 3.1 of the revised manuscript.

*P8ln15-16: saying that large errors could be attributable to KF and Morrison schemes is pretty*

*meaningless.*

**Response:** The convective precipitation dominated the overprediction of total precipitation in the southern oceanic area, which may be possibly due to overprediction of convective precipitation intensity by the Kain–Fritsch cumulus scheme. The non-convective precipitation dominated the overprediction of total precipitation in the northeastern oceanic area, which could be attributed to possible errors in the Morrison double-moment microphysics scheme. We have clarified this in the Section 3.1 of the revised manuscript.

*P8ln28: Are the results shown in Fig5 averages for all 5 years?*

**Response:** Yes, they are. We have clarified this in the Section 3.1 of the revised manuscript.

*P9ln16: what are "upper BCs"? and where do they come from? And why are they particularly uncertain?*

**Response:** Upper BCs represent upper layer boundary conditions (BCs) of $O_3$, which are derived from CESM. Because total column $O_3$ is mainly determined by $O_3$ concentrations in upper troposphere (Tang et al., 2009; Zhang et al., 2016a, c), the overpredictions of TOR (column $O_3$) can be largely attributed to the inappropriateness of the upper layer BCs of $O_3$. From the results of Tang et al. (2009), we could also find large uncertainties in upper layer BCs of $O_3$. We have clarified this in the Section 3.1 of the revised manuscript.

*P9ln16-17: Another meaningless statement about uncertainties in about everything possibly causing errors in NO2 column. Can you provide more insightful analyses?*

**Response:** Column $NO_2$ was moderately overpredicted by 18.3%. Potential uncertainties in $NO_x$ emissions and the model treatment of deposition and chemistry processes may contribute to the model-observation difference. As discussed by Lin et al. (2010) and Han et al. (2015), there are several uncertainties in the modeled $NO_x$ lifetime. Uncertainties in the $NO_2$ column retrievals from OMI (with a relative error of 25%, Boersma et al., 2011) and the averaging kernels (Han et al., 2015) could also help to explain the bias. We have added this in the Section 3.1 of the revised manuscript.

*P9ln28-30: Please clarify this sentence.*

**Response:** For the air quality application driven with NCEP-FNL data, the observation and simulation data pairs for surface meteorological variables against NCDC observational data were on an hourly basis. The high correlations for major meteorological variables in Table S3 indicated that the model showed good skills in hourly meteorological predictions, thus NCEP-FNL data were sufficient to support the air quality applications for hourly air quality predictions. We have clarified this in the Section 3.2 of the revised manuscript.

*P10ln23-24: If a figure is important enough to be discussed in the text (S2) it should be in the main paper and not in the supplement. The reader should not need to see the supplement to follow the*

*discussion.*

**Response:** We think that Figure S2 is not so important to be discussed in the text. So we have removed Figure S2 and the corresponding discussion in the revised manuscript.

*P11ln12-13: The names of the simulations are confusing. The "baseline" is NCEP_BASE_Imp but the sensitivity is NCEP_BASE which sounds more like it should be the base.*

**Response:** The names of the simulations have been changed from CESM_BASE_Imp, CESM_BASE_Imp_Sens, NCEP_BASE_Imp and NCEP_BASE to CESM_BASE, CESM_BASE _Sens, NCEP_BASE and NCEP_BASE_WoImp.

*P11ln25-26: How are the fixed BCs derived?*

**Response:** The fixed BCs are provided by the operational CMAQ system. We have clarified this in the Sections 2.1 and 3.3 of the revised manuscript.

*P12ln4: S4 should be in main paper.*

**Response:** As suggested, we have moved Figure S4 to the main paper (i.e., Figure 11 in the revised manuscript).

*P12ln15: "close" should be "closer"*

**Response:** Revised as suggested.

*P13ln11-12: Aerosol effects on photolysis in CMAQ do not depend on aerosol feedback in the WRF-CMAQ system. The more likely cause for ozone decline in the feedback run is increased NOx titration in cities due to reduced PBL mixing. Table 1: what LSM and surface layer scheme? Table 3 and 4: Better to have un-normalized error for T2, RH2, WS10, WD10*

**Response:** The decrease in $O_3$ concentrations in the feedback run may be attributed to the increased $NO_x$ titration resulted from increased atmospheric stability and reduced PBL height. We have clarified this in the Section 3.4 of the revised manuscript. Table 1: The land surface model is Pleim–Xiu land surface model (Xiu and Pleim, 2001). The surface layer scheme is Pleim–Xiu surface layer scheme. We have added them into Table 1. Table 3 and 4: We have added the mean absolute gross error (MAGE) in Table 3, and added the root mean square error (RMSE) in place of the normalized mean error (NME) in Tables 3 and 4.

**Anonymous Referee #2**

*This study reports the evaluation against measurements of the output from a dynamical downscaling link between the global Community Earth System Model (CESM) and the WRF-CMAQ modelling system over the East Asia region for a number of meteorological and air quality composition variables. The climatological simulations were for RCP4.5 for 2006-10 and the air quality applications were for winter and summer months in 2013 (principal compositional variables of interest: PM2.5 and O3). The authors report satisfactory prediction of major meteorological variables, although see the first of the general comments below. The paper reports on a major piece of work, with what appear to be generally appropriate methods, and is within the scope for consideration of publication in GMD.*

**Response:** We thank Referee #2 for the constructive comments. Please see below our point-by-point replies to other comments.

General comments
*(1) The description of the downscaling (P5-6) indicates that aspects of it involves significant bias corrections, so to what extent is it valid to judge model performance by model-observation statistics? For example, it is stated on P8, lines 1-6, that the improved statistical performance of the modelling approach used in this study may be related to the bias-correction applied. If a bias correction is applied then presumably we expect better model-observation statistics, so have we learned anything fundamental about the model performance by these comparison statistics?*

**Response:** While using bias-corrected ICs/BCs does improve WRF-CMAQ's model performance, it does not make model-observation comparison invalid. While meteorological reanalysis data were used to correct biases in meteorological ICs/BCs based on CESM-NCSU's results and satellite retrievals of $O_3$ were used to constrain their upper boundary conditions, observational data were used for model performance evaluation. Because GCMs generally suffer from systematic biases to a certain extent, bias correction to the GCM (i.e., CESM) boundary conditions was applied in this study to improve the model performance in simulating regional climate. By comparing to the traditional approach without GCM bias corrections, previous studies (Xu and Yang, 2012; Bruyère et al., 2014; Done et al., 2015) have shown that the improved dynamical downscaling method with GCM bias corrections greatly improves the downscaled climate. The bias-correction technique is also used in the NCAR CESM global bias-corrected CMIP5 output to support WRF/MPAS research (https://rda.ucar.edu/datasets/ds316.1/). Also note that the bias correction is applied to the ICs/BCs, rather than the model results. So, the model-observation comparison will provide insights into the model's capability in capturing observations.

*(2) The model-observation statistics should include RMSE instead of, or in place of, the normalized mean error (NME). The former is the statistic usually used alongside the correlation coefficient and mean bias (or normalised mean bias) in the suite of statistics that captures the spectrum of model performance characteristics.*

**Response:** As suggested, we have added the root mean square error (RMSE) in the statistics tables (Table 3, 4 and S3) in place of the normalized mean error (NME), and added the mean absolute gross

error (MAGE) in Table 3. The model performed well for T2 and RH2, with MBs of -0.6 ℃ and 0.8%, correlation coefficients of 0.97 and 0.72, MAGEs of 2.4 ℃ and 9.7%, and RMSEs of 3.2 ℃ and 12.6%, respectively. WS10 was moderately overpredicted by 22.2%, with an MB of 0.6 m/s, an MAGE of 1.2 m/s and a RMSE of 1.6 m/s. We have added this in the Section 3.1 of the revised manuscript.

*(3) In general, the discussion of model output against meteorological and compositional variations is (i) vague, i.e. non-quantitative (using phrasing like agreed well, satisfactory, etc.), and (ii) lacking explanatory insight, i.e. lists of potential reasons for discrepancy are given which could be written down as potential explanations without needing to do these comparisons. The authors should endeavour to provide more quantitative assessments of model performance, including how their mod-obs statistics compare with expectation and with other studies, and also to provide some more informed analysis of what is the driving explanation for mod-obs discrepancies for particular variables or circumstances.*

**Response:** As suggested, we have provided more quantitative assessments of model performance in terms of MB, NMB, or RMSE in the abstract, result and conclusion sections of the revised manuscript. The model biases or errors can be attributed to many factors. Pinpointing the exact causes is not a trivial effort, often involving large amounts of sensitivity simulations and in some cases, model further development and improvement that are not permitted with our very limited resources. Nevertheless, we have provided more insights into the model's performance statistics and how they are compared with other studies, wherever possible. For example, we have compared the performance of several meteorological variables with the benchmarks suggested by Emery et al. (2001) in the Section 3.1 of the manuscript. Emery et al. (2001) suggested the benchmarks for satisfactory performance for T2 (MB within ±0.5 ℃, MAGE of ≤ 2.0 ℃) and WS10 (MB within ±0.5 m/s, MAGE and RMSE of 2.0 m/s). In the climatological application, the MB and MAGE of T2 and the MB of WS10 are close to the benchmark, the MAGE and RMSE of WS10 are within the benchmark, and hence the performance is deemed acceptable.

We have also compared the CMAQ performance of chemical predictions in this study with other studies, as shown in the Section 3.2 of the manuscript. The revised text is as follows:

The CMAQ performance of chemical predictions in this study was comparable to or even better than those of other air quality studies over East Asia (Wang et al., 2009; 2012; Liu et al., 2010; Zheng et al., 2015; Hu et al., 2016; Liu et al., 2016; Zhang et al., 2016a). This study predicted relatively well for most chemical species in most months. Compared with other regional modeling studies, WRF-CMAQv5.0.2 used in this study outperformed MM5/CMAQv4.6, which tend to underpredict the surface concentrations of major species with NMBs generally greater than -40% and overpredict surface $O_3$ concentrations in most months with NMBs generally higher than 20% over East Asia according to the evaluation results of Zhang et al. (2016a). A relatively good performance of CMAQv5.0.1 was also reported by Hu et al. (2016). Global models such as GEOS-Chem and CESM tend to underpredict $PM_{2.5}$ concentrations (by about -50% as reported by Jiang et al., 2013) and overpredict $O_3$ concentrations (by about 50% as reported by He and Zhang, 2014; Wang et al., 2013) in China/East Asia because of relatively coarse grid resolution and limitations in some model treatments (e.g., missing emissions of unspeciated primary $PM_{2.5}$, and discrepancies in surface layer height and vertical mixing).

Specific comments

*P1, L27: The phrasing "The model showed good ability to predict PM2.5 . . ..and O3. . ." is non-quantitative and vague.*

**Response:** The above sentence has been revised to include more quantitative assessment as follows:

The model showed good ability to predict $PM_{2.5}$ in winter (with a normalized mean bias (NMB) of 6.4% in 2013) and $O_3$ in summer (with an NMB of 18.2% in 2013) in terms of statistical performance and spatial distributions.

In addition, we have added this in the abstract of the revised manuscript.

*P4, L20: Rephrase as "Several modifications in model. . ."*

**Response:** Revised as suggested.

*P7, L16: Although the acronym TOR is defined here, there needs to be some further explanation of what it means in practice, particularly in the context of its relevance to model performance evaluation.*

**Response:** TOR represents tropospheric ozone residual or column abundance of $O_3$. We have clarified this in the Section 2.2 of the revised manuscript.

*P12, L18: "were much closer to. . .*

**Response:** Revised as suggested.

**Reference**

[revised manuscript text omitted]

**Supplement**

**Configuration used in CESM-NCSU simulations**

Table S1 summarizes The CESM-NCSU configurations for simulations under the RCP4.5 scenario. More detailed descriptions can be found in He and Zhang (2014) and Glotfelty et al. (2017a, b).

**Table S1.** The CESM-NCSU configurations for simulations under the RCP4.5 scenario.

| Attribute or Process | Configuration |
|---|---|
| Simulation Time Period | Current decade (2001-2010) and future decade (2046-2055) |
| Horizontal Resolution | $0.9° \times 1.25°$, 192 (latitudes) $\times$ 288 (longitudes) |
| Vertical Resolution | 30 layers from 1000 mb to 3 mb |
| Deep Convection | Zhang and McFarlane (1995); Neale et al. (2008) |
| Shallow Convection | Park and Bretherton (2009) |
| Cloud Microphysics | Morrison and Gettelman (2008) |
| Planetary Boundary Layer | Bretherton and Park (2009) |
| Short and Long-wave Radiation | RRTMG (Iacono et al., 2003, 2008) |
| Gas-phase Chemistry | CB05GE (Karamchandani et al., 2012) |
| Aqueous Chemistry | Barth et al. (2000) |
| Aerosol Module | Modified MAM7 (Liu et al., 2012; He and Zhang, 2014) |
|    Inorganic Aerosol Thermodynamics | ISORROPIA II (Fountoukis and Nenes, 2007) |
|    VBS secondary organic aerosol model | Glotfelty et al. (2017b) |
| Aerosol Activation | Fountoukis and Nenes (2005); Barahona et al. (2010); Kumar et al. (2009) |

RRTMG: Rapid Radiative Transfer Model for General Circulation Models; CB05GE: Carbon Bond Mechanism 2005 with Global Extension; MAM7: Modal Aerosol Model with Seven modes; VBS: Volatility Basis Set.

**Mapping between CESM/CAM5 and CMAQ aerosol species**

The mapping table between CESM/CAM5 and CMAQ aerosol species is shown in Table S2. The CESM/CAM5 uses the 7-mode prognostic Modal Aerosol Model (MAM7) (Liu et al., 2012) with volatility-basis-set (VBS) (Glotfelty et al., 2017b), whereas CMAQ uses the 3-mode AERO6 aerosol module. The MAM7 in CESM/CAM5 includes Aitken (2), accumulation (1), primary carbon (3), fine dust (5), fine sea salt (4), coarse dust (7) and coarse sea salt (6) modes. The AERO6 in CMAQ includes Aitken (I), accumulation (J) and coarse (K) modes, which is similar to MAM3 (Liu et al., 2012). Similar to the mapping of aerosol modes between MAM7 and MAM3 in Liu et al. (2012), the Aitken mode in MAM7 is mapping to the Aitken mode (I) in AERO6; the accumulation, primary carbon, fine dust and fine sea salt modes in MAM7 are mapping to the accumulation mode (J) in AERO6; the coarse dust and coarse sea salt modes in MAM7 are mapping to the coarse mode (K) in AERO6. For example, sulfate in accumulation mode (so4_a1), fine sea salt mode (so4_a4) and fine dust mode (so4_a5) in MAM7 are mapping to sulfate in accumulation mode (ASO4J) in AERO6.

Secondary organic aerosol (SOA) species in CESM/CAM5 were divided according to different volatility levels. However, the CMAQ model includes specific SOA semi-volatile and nonvolatile species. The anthropogenic and biogenic SOA species in CESM/CAM5 were first lumped into total semi-volatile SOA and total nonvolatile SOA. The ratios among the SOA species derived from the default BCs/ICs were then used to allocate each SOA species in CMAQ based on the combined SOA, as suggested by Carlton et al. (2010).

1    **Table S2.** Mapping table between CESM/CAM5 and CMAQ aerosol species.

| CMAQ | CESM/CAM5 |
|---|---|
| J - Accumulation | 1 - Accumulation |
| I - Aitken | 2 - Aitken |
| J - Accumulation | 3 - Primary Carbon |
| J - Accumulation | 4 - Fine Sea Salt |
| J - Accumulation | 5 - Fine Dust |
| K - Coarse | 6 - Coarse Sea Salt |
| K - Coarse | 7 - Coarse Dust |
| ASO4J | so4_a1+so4_a4+so4_a5 |
| ASO4I | so4_a2 |
| ASO4K | so4_a6+so4_a7 |
| ANO3J | no3_a1+no3_a4+no3_a5 |
| ANO3I | no3_a2 |
| ANO3K | no3_a6+no3_a7 |
| ANH4J | nh4_a1+nh4_a4+nh4_a5 |
| ANH4I | nh4_a2 |
| ANH4K | nh4_a6+nh4_a7 |
| AECJ+AECI | bc_a1+bc_a3 |
| APOCJ+APNCOMJ+APOCI+APNCOMI | poa1_a1+poa2_a1+poa3_a1+poa4_a1+poa5_a1+poa6_a1+poa7_a1+poa1_a3+poa2_a3+poa3_a3+poa4_a3+poa5_a3+poa6_a3+poa7_a3 |
| AALKJ+AXYL1J+AXYL2J+ATOL1J+ATOL2J+ABNZ1J+ABNZ2J | asoa2_a1+asoa2_a2+asoa3_a1+asoa3_a2+asoa4_a1+asoa4_a2 |
| AXYL3J+ATOL3J+ABNZ3J+AOLGAJ | asoa1_a1+asoa1_a2 |
| ATRP1J+ATRP2J+AISO1J+AISO2J+ASQTJ | bsoa2_a1+bsoa2_a2+bsoa3_a1+bsoa3_a2+bsoa4_a1+bsoa4_a2 |
| AISO3J+AOLGBJ | bsoa1_a1+bsoa1_a2 |
| AORGCJ | soa_a1+soa_a2 |
| ANAJ | na_a1+na_a4+na_a2 |
| ASEACAT | na_a6 |
| ACLJ | cl_a1+cl_a4+cl_a5 |
| ACLI | cl_a2 |
| ACLK | cl_a6+cl_a7 |
| AOTHRJ+AFEJ+AALJ+ASIJ+ATIJ+ACAJ+AMGJ+AKJ+AMNJ | dst_a5 |
| ACORS+ASOIL | dst_a7 |

**Evaluation of dust simulation in CESM-NCSU**

The 5-year average (2006-2010) $PM_{10}$ concentrations from CESM-NCSU were evaluated by comparison with observed data in 2013 to assess the performance of the dust emission scheme used in CESM-NCSU. CESM-NCSU tends to overpredict dust concentrations over East Asia in April, and a scale factor of 1/3 was thus applied to adjust dust concentrations from CESM-NCSU, which helped reduce the high bias in dust simulation (see Fig. S1).

[Figure]

(a)                                    (b)

**Fig. S1.** 5-year average (2006-2010) simulated $PM_{10}$ concentrations in April from (a) original CESM-NCSU and (b) dust-revised CESM-NCSU (CESM_0.33Dust) overlaid with observations in 2013.

1 **Table S3.** Model performance statistics for the air quality application: meteorological variables (2013,

2 NCEP_BASE).

| Variable | Network | January | | | | April | | | | July | | | |
|---|---|---|---|---|---|---|---|---|---|---|---|---|---|
| | | R | MB | NMB (%) | RMSE | R | MB | NMB (%) | RMSE | R | MB | NMB (%) | RMSE |
| T2 (°C) | NCDC | 1.0 | 0.2 | -105.4 | 3.8 | 0.9 | -1.2 | -10.1 | 3.5 | 0.8 | -1.8 | -7.3 | 3.6 |
| RH2 (%) | NCDC | 0.6 | 4.0 | 5.9 | 17.5 | 0.7 | 3.4 | 5.4 | 17.9 | 0.7 | 2.8 | 3.7 | 14.7 |
| WS10 (m s$^{-1}$) | NCDC | 0.6 | 0.7 | 26.3 | 2.3 | 0.6 | 0.2 | 7.0 | 2.2 | 0.5 | 0.2 | 6.3 | 1.9 |
| WDR10 (degree) | NCDC | 0.4 | 7.4 | 3.6 | 124.8 | 0.4 | 4.4 | 2.2 | 107.2 | 0.3 | 5.9 | 3.2 | 94.4 |
| Precip (mm day$^{-1}$) | NCDC | 0.1 | 0.3 | 35.4 | 5.3 | 0.5 | 0.2 | 7.7 | 6.9 | 0.4 | 0.4 | 7.7 | 14.4 |
| Precip (mm day$^{-1}$) | GPCP | 0.7 | -0.2 | -16.9 | 1.2 | 0.7 | -0.4 | -21.3 | 1.6 | 0.7 | -0.4 | -6.8 | 4.5 |
| SWDOWN (W m$^{-2}$) | CERES | 0.9 | 13.5 | 11.1 | 23.1 | 0.8 | 33.1 | 14.4 | 41.1 | 0.7 | 42.6 | 18.9 | 56.4 |
| LWDOWN (W m$^{-2}$) | CERES | 1.0 | -9.8 | -3.6 | 16.4 | 1.0 | -14.3 | -4.4 | 18.7 | 1.0 | -11.6 | -3.0 | 18.8 |
| GSW (W m$^{-2}$) | CERES | 0.9 | 2.3 | 2.4 | 20.1 | 0.8 | 18.2 | 9.4 | 30.9 | 0.7 | 30.7 | 15.6 | 45.0 |
| OLR (W m$^{-2}$) | CERES | 1.0 | 3.0 | 1.3 | 10.4 | 0.9 | 5.9 | 2.4 | 13.6 | 0.7 | 5.3 | 2.3 | 23.2 |
| SWCF (W m$^{-2}$) | CERES | 0.8 | 4.5 | -16.1 | 16.7 | 0.8 | 20.2 | -38.1 | 27.0 | 0.7 | 22.1 | -26.8 | 38.6 |
| LWCF (W m$^{-2}$) | CERES | 0.6 | -6.8 | -41.6 | 11.1 | 0.6 | -11.5 | -42.8 | 15.5 | 0.6 | -11.5 | -25.5 | 24.0 |
| CF (%) | MODIS | 0.6 | -23.5 | -34.2 | 33.1 | 0.5 | -19.2 | -31.4 | 28.9 | 0.5 | -17.4 | -23.8 | 30.2 |

3 [1] R: correlation coefficient; MB: mean bias; NMB: normalized mean biases; RMSE: root mean square error.

[Figure]

2 **Fig. S2.** O$_3$ boundary conditions (BCs) in January derived from (a) CESM and (b) fixed boundary

3 conditions (BCs).

---

## Author Response (AR2)

**Topical Editor**

*Comments to the Author:*

*Dear authors,*

*First of all, I would like to acknowledge the two reviewers for their constructive feedback on your original manuscript. I would also like to thank you for addressing their comments and for submitting a revised manuscript.*

*On reading your responses to the reviewers, I feel that you have largely addressed their comments. However, I have the following comments that I would like you to clarify. Once these are addressed, I will be very pleased to recommend that your manuscript be accepted for publication in GMD.*

*Regards,*

*Fiona O'Connor*

**Response:** We would like to acknowledge the editor and the two reviewers for their constructive comments. We address the editor's comments as below.

*Comments*

*1. Reviewer 1 - My primary criticism of this paper is the lack of detailed description of the GCM model, the downscaling, regional model configuration, and execution. Even though references are given for the CESM modeling and the chemical and aerosol processes are briefly described I would like to see further description of the CESM physics, spin-up, constraints, etc. I do not understand how this represents a climate scenario when it is for past years and is evaluated against observations. What does RCP4.5 for these years represent. Do these runs use observation based SSTs? If these runs were spun-up from pre-industrial times without any observed data constraints, there would be no reason to expect agreement with observations. If bias corrections are made to both the meteorology and chemistry, then how do these runs substantially differ from re-analysis driven runs? Please explain the rationale and expectations of these runs.*

*There are essentially 2 questions here. The first is related to the model and downscaling descriptions and the second seems to arise from a confusion about the use of RCP4.5 to model the time period 2006-2010. You've very clearly set out the rationale for using RCP4.5 although I ask for some further clarification on a related comment by the same reviewer - see below.*

*In answer to the first question, I'm a firm believer that a paper should stand alone in its own right and that the reader shouldn't be expected to look for basic information in other papers. However, I think there is also a balance to be struck between the level of detail that one puts in a paper and the extent to which you cite the pre-existing literature. Having said that, I think your response to this comment is appropriate, adding in more detail on the GCM, downscaling etc. (Section 2.2. and Table S1). However, I have two minor comments related to the newly added text that I would like you to address:*

*(i) Why are the ice and the ocean models initialised using CESM default settings rather than using output from an appropriate simulation of CESM run for CMIP5?*

*(ii) There's a bit too much repetition in the added text in Section 2.2, particularly lines 22-24 i.e. The CESM simulation … fully-coupled CESM .. with CESM configuration… fully-coupled CESM …*

**Response:** (i) We indeed initialized future-year simulations based on CMIP5 simulations. Only the current year simulations used the default ocean and ice settings. This is mainly to maintain consistency

with the 10 current-year simulations that were previously performed using more realistic emissions (emissions based on EDGAR and regional emission inventories) such as NEI from US EPA (see papers published by He and Zhang, 2014, He et al., 2015a in the reference list) so that we can intercompare the 10 current-year results using RCP emissions (Glotfelty et al., 2017a) that were used in this work against those using more realistic emissions that were performed previously.

(ii) The sentence has been revised as follows: The CESM simulation for 2001–2010 is performed with fully coupled CESM1.2.2 with the B_2000_STRATMAM7_CN configuration, which includes prognostic simulation of the atmosphere, ocean, land, and sea ice from the various component models.

*2.    Reviewer 1 - P4lns21-22: This statement about "correcting the roughness length by increasing the friction velocity by 1.5 times when calculating wind speeds in the ACM2 PBL scheme to reduce the overpredictions of wind speeds" needs more explanation. First, if the roughness lengths need correcting why not change them and not the friction velocity. Second, what is the problem with roughness lengths? How are they specified and what are they? Our experience has not shown general overpredictions in windspeed. Windspeed and friction velocity are strongly affected by the LSM and surface layer scheme which are not even mentioned here. Also the LU scheme and data are important. The USGS 24cat data is way out of date especially for China where urbanization has been dramatic. Why not use MODIS LU?*

*I thought that you addressed these comments very well but I would like to see you use more of your justification for scaling the friction velocity included in Section 2.1.*

**Response:** As suggested, we have added our justification for scaling the friction velocity in the Section 2.1 of the revised manuscript. The revised text is as follows:

These included (1) correcting the surface roughness by increasing the surface drag (which is applied to the friction velocity) by 1.5 times when calculating wind speeds in the ACM2 PBL scheme to reduce the overpredictions of wind speeds, which are likely caused by low surface drag due to the inappropriate representation of surface roughness because the detailed surface structure cannot be reproduced at a coarse grid resolution of 36-km (Mass and Ovens, 2010; Zheng, et al., 2015; Zhang et al., 2016b).

*3.    Reviewer 1 - Page 6: I don't understand what is the point of using RCP projections when modeling retrospectively. It seems that 2008 emission inventories are used for more detailed spatial-temporal allocation. Then why not just use these inventories? What is an RCP projection for past years? Please explain the logic here*

*Can you clearly add somewhere that the CMIP5 historical emissions (Lamarque et al., ACP, 2010) only covered up to 2005 and that the RCP emissions covered the period 2005-2100. Then it will be clearer why you are using RCP4.5 for those sectors not included in MIX2008.*

**Response:** The RCP dataset provides global emission projections for the period 2005 to 2100 with identical base year 2000. We have clarified this in the Section 2.3 of the revised manuscript.

*4.    Reviewer 1 - P8ln28: Are the results shown in Fig5 averages for all 5 years?*

*Can you be explicit about this in the figure caption i.e. multi-annual annual means of CF,AOD etc.. for the time period 2006-2010*

**Response:** Revised as suggested.

*5.    Reviewer 2 - (3) In general, the discussion of model output against meteorological and compositional variations is (i) vague, i.e. non-quantitative (using phrasing like agreed well, satisfactory, etc.), and (ii) lacking explanatory insight, i.e. lists of potential reasons for discrepancy are given which could be written down as potential explanations without needing to do these comparisons. The authors should endeavour to provide more quantitative assessments of model performance, including how their mod-obs statistics compare with expectation and with other studies, and also to provide some more informed analysis of what is the driving explanation for mod-obs discrepancies for particular variables or circumstances.*
*I think you've addressed this quite well, by adding in the commonly used root mean square error as part of your statistical assessment as well as placing the WRF-CMAQ results in the context of other model results. Can you correct line 10 on page 12 i.e. GEOS-Chem and CESM tend to under-predict PM2.5 concentrations (by about -50%...). If the term "under-predict" is used, then there is no need to use -50% ... it should be 50%*

**Response:** Corrected as suggested.

**Reference**

[revised manuscript text omitted]